# Pseudohypoxic HIF pathway activation dysregulates collagen structure-function in human lung fibrosis

Christopher J Brereton[1,2†], Liudi Yao[3†], Elizabeth R Davies[1,2,3], Yilu Zhou[3,4], Milica Vukmirovic[5,6], Joseph A Bell[1,2], Siyuan Wang[3], Robert A Ridley[1,2], Lareb SN Dean[1,2], Orestis G Andriotis[7], Franco Conforti[1,2], Lennart Brewitz[8], Soran Mohammed[9], Timothy Wallis[1,2], Ali Tavassoli[9], Rob M Ewing[3,4], Aiman Alzetani[2,10], Benjamin G Marshall[2,10], Sophie V Fletcher[2,10], Philipp J Thurner[7], Aurelie Fabre[11], Naftali Kaminski[5], Luca Richeldi[2,12], Atul Bhaskar[13], Christopher J Schofield[8], Matthew Loxham[1,2,4], Donna E Davies[1,2,4], Yihua Wang[2,3,4*], Mark G Jones[1,2,4*]

[1]Clinical and Experimental Sciences, Faculty of Medicine, University of Southampton, Southampton, United Kingdom; [2]NIHR Southampton Biomedical Research Centre, University Hospital Southampton, Southampton, United Kingdom; [3]Biological Sciences, Faculty of Environmental and Life Sciences, University of Southampton, Southampton, United Kingdom; [4]Institute for Life Sciences, University of Southampton, Southampton, United Kingdom; [5]Section of Pulmonary, Critical Care and Sleep Medicine, Department of Medicine, Yale University School of Medicine, New Haven, United States; [6]Leslie Dan Faculty of Pharmacy, University of Toronto, Toronto, Canada; [7]Institute of Lightweight Design and Structural Biomechanics, TU Wien, Vienna, Austria; [8]Department of Chemistry and the Ineos Oxford Institute for Antimicrobial Research, Chemistry Research Laboratory, Oxford, United Kingdom; [9]School of Chemistry, University of Southampton, Southampton, United Kingdom; [10]University Hospital Southampton, Southampton, United Kingdom; [11]Department of Histopathology, St. Vincent's University Hospital & UCD School of Medicine, University College Dublin, Dublin, Ireland; [12]Unità Operativa Complessa di Pneumologia, Università Cattolica del Sacro Cuore, Fondazione Policlinico A. Gemelli IRCCS, Rome, Italy; [13]Faculty of Engineering and Physical Sciences, University of Southampton, Southampton, United Kingdom

*For correspondence:
yihua.wang@soton.ac.uk (YW);
mark.jones@soton.ac.uk (MGJ)

†These authors contributed equally to this work

Competing interest: The authors declare that no competing interests exist.

**Abstract** Extracellular matrix (ECM) stiffening with downstream activation of mechanosensitive pathways is strongly implicated in fibrosis. We previously reported that altered collagen nanoarchitecture is a key determinant of pathogenetic ECM structure-function in human fibrosis (Jones et al., 2018). Here, through human tissue, bioinformatic and ex vivo studies we provide evidence that hypoxia-inducible factor (HIF) pathway activation is a critical pathway for this process regardless of the oxygen status (pseudohypoxia). Whilst TGFβ increased the rate of fibrillar collagen synthesis, HIF pathway activation was required to dysregulate post-translational modification of fibrillar collagen, promoting pyridinoline cross-linking, altering collagen nanostructure, and increasing tissue stiffness. In vitro, knockdown of Factor Inhibiting HIF (FIH), which modulates HIF activity, or oxidative stress caused pseudohypoxic HIF activation in the normal fibroblasts. By contrast, endogenous FIH activity was reduced in fibroblasts from patients with lung fibrosis in association with significantly increased normoxic HIF pathway activation. In human lung fibrosis tissue, HIF-mediated signalling was increased at sites of active fibrogenesis whilst subpopulations of human lung fibrosis mesenchymal

cells had increases in both HIF and oxidative stress scores. Our data demonstrate that oxidative stress can drive pseudohypoxic HIF pathway activation which is a critical regulator of pathogenetic collagen structure-function in fibrosis.

## Editor's evaluation

The reviewers found your manuscript of broad interest to researchers interested in lung biology, as the study builds upon the previous original work of the authors, by identifying a pathway that regulates collagen nanostructure and stiffness in lung fibrosis and demonstrating that this pathway it is independent of pathways regulating collagen synthesis. They also valued the elegant analysis you performed to validate the specificity of experimental finding, and demonstrate that HIF activation is required for the increased tissue stiffness associated with fibrosis.

## Introduction

We previously identified that in the lung tissue of patients with idiopathic pulmonary fibrosis (IPF) there is increased pyridinoline collagen cross-linking and altered collagen fibril nano-architecture, with individual collagen fibrils being structurally and functionally abnormal (*Jones et al., 2018*). This was associated with increased tissue expression of lysyl hydroxylase 2 (LH2/PLOD2, which catalyses telopeptide lysine hydroxylation to determine pyridinoline cross-linking) and the lysyl oxidase-like (LOXL) enzymes LOXL2 and LOXL3, which initiate covalent collagen cross-linking (*Jones et al., 2018*). This pyridinoline cross-linking, rather than any change in collagen deposition per se, determined increased IPF tissue stiffness. Inhibiting pyridinoline cross-linking normalised mechano-homeostasis and limited the self-sustaining effects of ECM on fibrosis progression. Whilst identifying the importance of altered collagen nanoarchitecture to human lung fibrosis pathogenesis, the upstream mechanisms that dysregulate collagen structure-function to promote progressive fibrosis rather than tissue repair were not determined. Here, we investigated possible mechanisms and established their relevance to human lung fibrosis.

## Results

### The pyridinoline collagen fibrillogenesis genes PLOD2 and LOXL2 are co-expressed at sites of active fibrogenesis

In our previous work comparing human IPF lung tissue with age-matched control lung tissue, we identified that in bulk IPF lung tissue lysates there are significant increases in the relative expression levels of the collagen modifying enzymes *LOXL2*, *LOXL3*, and *LOXL4*, as well as *PLOD2* (also known as lysyl hydroxylase or LH2) (*Jones et al., 2018*). To further investigate this observation, we first studied the transcriptomic profiles of fibroblast foci, the sites of active fibrogenesis in IPF. We analysed a data set we recently generated by integrating laser-capture-microdissection and RNA-Seq (LCMD/RNA-seq) which enabled profiling of the in situ transcriptome of fibroblast foci as well as alveolar septae from control tissue and IPF tissue (Gene Expression Omnibus (GEO) GSE169500). The LOXL enzyme with the greatest expression in fibroblast foci was *LOXL2* (*Figure 1a–e*). Whilst *LOXL3* and *LOXL4* expression was increased within fibroblast foci, only limited expression was identified (*Figure 1d and e*). Furthermore, *PLOD2* expression was significantly increased within fibroblast foci (*Figure 1f*), and *PLOD2* expression levels correlated ($r = 0.63$, $P = 0.04$) with those of *LOXL2* (*Figure 1g*) but not with those of other LOXL enzymes (*Figure 1—figure supplement 1a-d*), suggesting possible co-ordinated regulation of *PLOD2* and *LOXL2* gene expression. By contrast, expression of the major collagen fibrillogenesis gene *COL1A1* did not significantly correlate with their expression (*Figure 1—figure supplement 1e*), suggesting that, in lung fibrosis, distinct pathways might promote pyridinoline cross-linking to dysregulate collagen fibril nano-structure independently of pathways regulating major fibrillar collagen synthesis. We then performed RNA in situ hybridisation upon IPF lung tissue, with semi-quantitative analysis (*Supplementary file 1a*); the results showed that the greatest expression of *LOXL2* and *PLOD2* in IPF tissue was by mesenchymal cells within fibroblast foci, and that *LOXL2* and

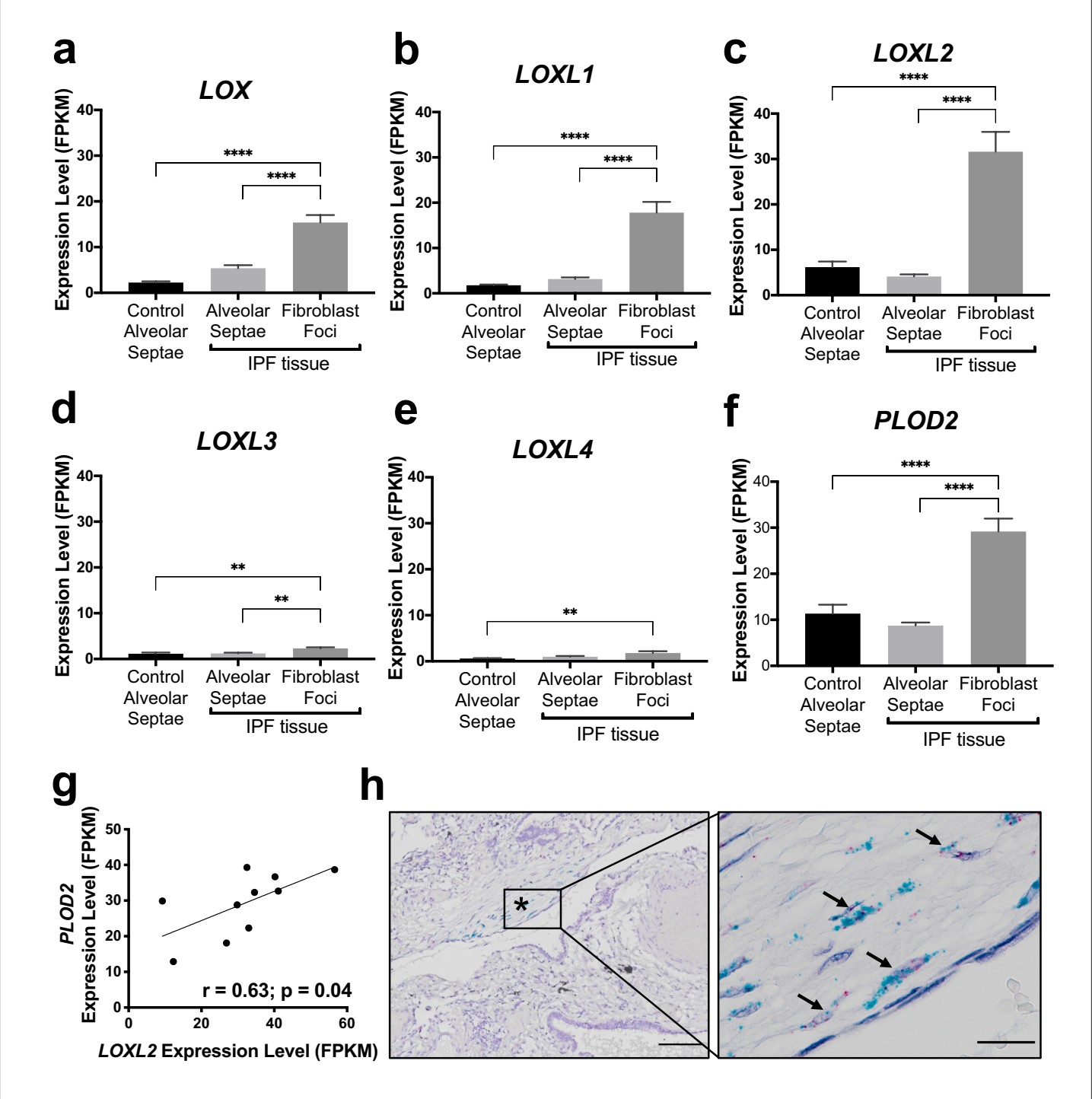

**Figure 1.** The collagen cross-linking enzymes PLOD2 and LOXL2 are co-expressed at sites of active fibrogenesis in IPF. (**A–F**) Expression of *LOX*, *LOXL1*, *LOXL2*, *LOXL3*, *LOXL4,* and *PLOD2* in healthy alveolar septae, IPF alveolar septae and IPF fibroblast foci (n = 10 individual healthy and IPF donors). Relative expression levels are calculated as Fragments Per Kilobase of transcript per Million mapped reads (FPKM). Bars represent standard geometric means. **p < 0.01; ****p < 0.0001 by Tukey's multiple comparisons test. (**G**) Scatterplot of paired fibroblast foci data from (**C**) and (**F**) were plotted to compare expression of PLOD2 and LOXL2 (Spearman rank correlation coefficient r = 0.63, p = 0.04). (**H**) Representative image of mRNA expression of *PLOD2* (red chromagen) and *LOXL2* (green chromagen) in IPF lung tissue (n = 7 donors) using RNAscope RNA in-situ hybridisation. A fibroblastic focus is identified by * and arrows identify co-expression pattern. Left scale bar 100 µm, right scale bar 20 µm.

The online version of this article includes the following figure supplement(s) for figure 1:

**Figure supplement 1.** Correlation of PLOD2 with LOXL family members.

*PLOD2* are co-expressed within the same cells (*Figure 1h*) within areas of fibrillar collagen deposition (*Figure 1—figure supplement 1f-h*).

## HIF pathway activation is a key inducer of PLOD2 and LOXL2 expression in lung fibroblasts

To investigate common regulators of *PLOD2* and *LOXL2* in lung fibrosis, we studied their expression in primary human lung fibroblasts over a 72-hr time course following activation (*Figure 2—figure supplement 1a*) of transforming growth factor beta (TGFβ), epidermal growth factor (EGF), hypoxia inducible factors (HIF) or Wnt signalling pathways, each of which have been implicated in fibrogenesis (*Richeldi et al., 2017*; *Yao et al., 2019*; *Martin-Medina et al., 2018*; *Königshoff et al., 2008*; *Yue et al., 2010*; *Bodempudi et al., 2014*; *Hill et al., 2019a*; *Yao et al., 2021*; *Zhou et al., 2021*). A prodrug form of the hypoxia mimetic and broad spectrum 2-oxoglutarate oxygenase inhibitor N-oxalylglycine (dimethyloxalylglycine, DMOG) (*Chowdhury et al., 2013*), which inhibits the HIF prolyl hydroxylases with consequent stabilisation of HIF1α and HIF2α, most strongly upregulated both PLOD2 and LOXL2 mRNA and protein levels (*Figure 2a–c* and *Figure 2—figure supplement 1b*) but did not induce expression of interstitial collagen genes (*COL1A1*, *COL3A1*) (*Figure 2d* and *Figure 2—figure supplement 1c*). In contrast, TGFβ1 strongly induced *COL1A1* and *COL3A1* and this was associated with smaller up-regulation of *PLOD2* at 24 hr and of *LOXL2* at 72 hr (*Figure 2a–d*; *Figure 2—figure supplement 1c*). No induction of *PLOD2* or *LOXL2* was identified with canonical Wnt (Wnt3a), non-canonical Wnt (Wnt5a) or EGF pathway activation (*Figure 2a–c*). We further extended these observations by showing that treatment with the selective HIF prolyl 2 hydroxylase inhibitor, *N*-[[1,2-dihydro-4-hydroxy-2-oxo-1-(phenylmethyl)–3-quinolinyl]carbonyl]-glycine (IOX2) (*Chowdhury et al., 2013*) or culture for 24 hr under hypoxic conditions induced expression of *PLOD2* and *LOXL2* (*Figure 2—figure supplement 1d, e*), with immunofluorescent staining confirming an increase in intracellular LOXL2 and PLOD2 expression following DMOG or IOX2 treatment in comparison to treatment with TGFβ1 (*Figure 2e*). Transcriptional activation of HIF pathways requires assembly of a heterodimer between HIF1α or HIF2α and their obligate binding partner HIF1β (*Schödel and Ratcliffe, 2019*; *Schofield and Ratcliffe, 2004*). To confirm the dependence of the induction of PLOD2 and LOXL2 expression upon HIF levels, siRNA knockdown against HIF1α(*HIF1A*), HIF2α(*EPAS1*), and HIF1β (*ARNT*) was performed (*Figure 3a*). The knockdown of HIF1α, but not HIF2α prevented DMOG induction of *PLOD2* mRNA and protein expression, whilst LOXL2 required silencing of both HIF1α and HIF2α or HIF1β (*Figure 3b–d*). Together, these findings identify that HIF stabilisation is required to orchestrate induction of *PLOD2* and *LOXL2* expression in human lung fibroblasts.

## HIF pathway activation and TGFβ1 synergistically increase PLOD2 expression

Given that TGFβ1 strongly induced major collagen fibrillogenesis genes whilst HIF pathways most strongly increased PLOD2 and LOXL2 expression levels, we investigated the effects of activating these pathways individually or in combination using lung fibroblasts from patients with IPF. The effect of DMOG in the absence or presence of TGFβ1 upon PLOD2 and LOXL2 induction (*Figure 4a–c*) was comparable to that identified using normal control lung fibroblasts. When combined, a synergistic effect upon the induction of PLOD2 expression was apparent which was greater than either pathway alone (*Figure 4a and c*). Whilst expression of *LOXL2* was also increased with the combination of HIF stabilisation and TGFβ1, a corresponding increase in LOXL2 protein levels within cell lysates was not apparent. As LOXL2 is processed intracellularly before being extracellularly secreted, we therefore investigated whether increased secretion of LOXL2 was occurring; this identified that under conditions with HIF stabilisation LOXL2 secretion was increased in both IPF fibroblasts (*Figure 4d*) and control fibroblasts (*Figure 4—figure supplement 1*).

Although TGFβ1 alone was sufficient to induce interstitial collagen gene (*COL1A1*) expression (*Figure 4—figure supplement 2*), HIF stabilisation significantly increased the ratio of *PLOD2* and *LOXL2* gene expression relative to fibrillar collagen (*COL1A1*) gene expression while TGFβ1 did not (*Figure 4e and f*), suggesting that TGFβ activity alone may be insufficient to promote the altered collagen cross-linking that is present in IPF lung tissue. Together these findings demonstrate that whilst TGFβ1 has a dominant role in increasing the rate of synthesis of major fibrillar collagens, HIF

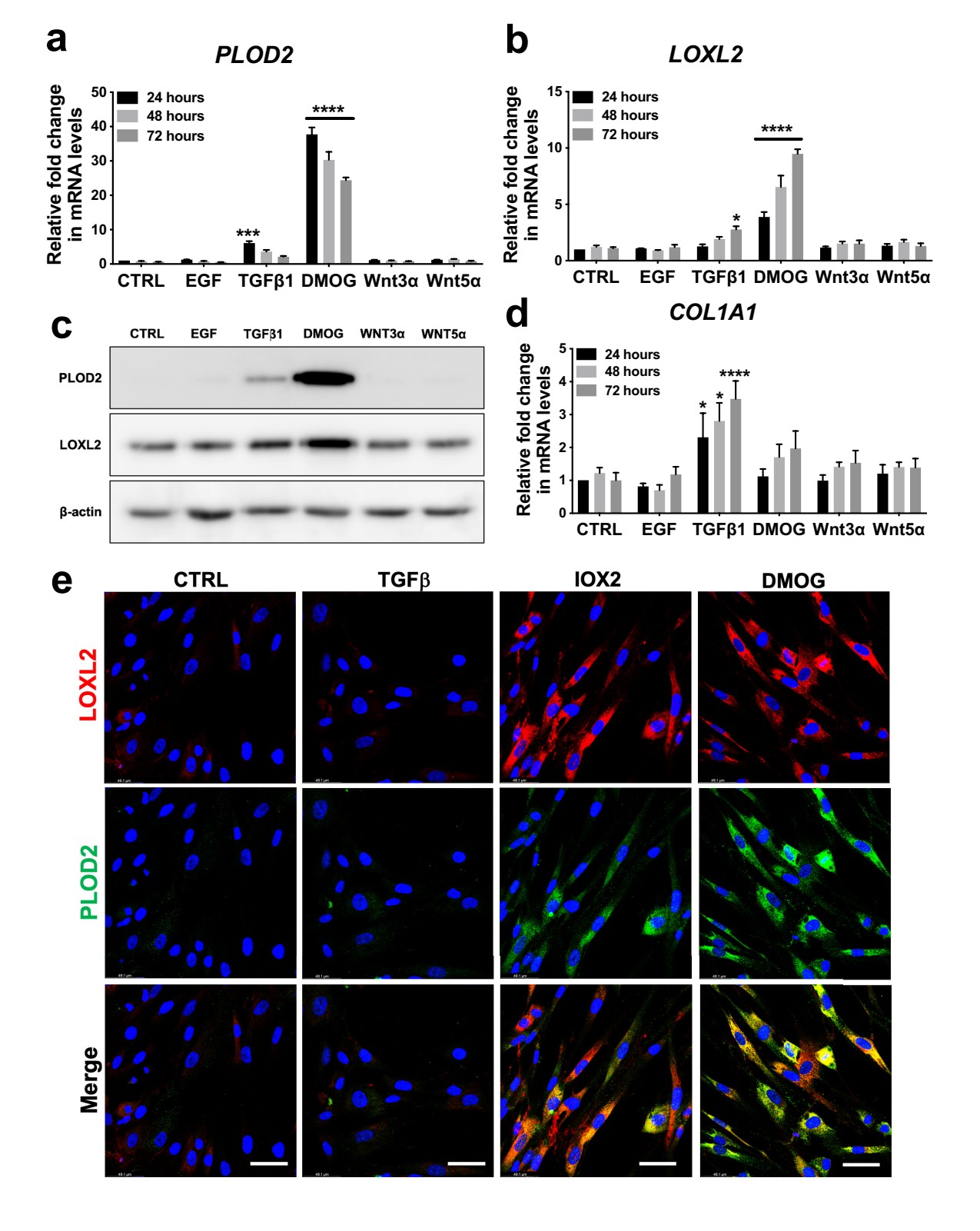

**Figure 2.** Hypoxia mimetics strongly promote PLOD2 and LOXL2 expression in lung fibroblasts. (**A–B, D**) Relative gene expression using the ΔΔCt method of *PLOD2*, *LOXL2,* and *COL1A1* in healthy lung fibroblasts over a 72-hr time course in the presence of EGF, TGFβ1, the hypoxia mimetic DMOG, Wnt3a, Wnt5a, or vehicle control. n = 3 independent experiments. Bars indicate geometric means. *p < 0.05; ***p < 0.001; ****p < 0.0001 by Dunnett's multiple comparisons test. (**C**) PLOD2 and LOXL2 protein levels at 72 hr. β-actin loading control. The full blots are shown in *Figure 2—source*

*Figure 2 continued on next page*

*Figure 2 continued*

**data 1**. (**E**) Representative immunofluorescence images of healthy lung fibroblasts with indicated treatment stained for LOXL2 (red), PLOD2 (green), and DAPI (blue). Scale bar 50 μm.

The online version of this article includes the following source data and figure supplement(s) for figure 2:

**Source data 1.** Full membrane scans for western blot images for *Figure 2c*.

**Figure supplement 1.** Pro-fibrotic signalling pathways in human lung fibroblasts.

**Figure supplement 1—source data 1.** Full membrane scans for western blot images for *Figure 2—figure supplement 1a, b, d*.

pathways may have a key role in regulating pathological post-translational modifications and collagen structure in lung fibrosis.

## HIF pathway activation alters collagen structure-function and increases tissue stiffness

To investigate whether HIF pathway activation acts as a mechanism that drives pathologic collagen crosslinking by disproportionate induction of collagen-modifying enzymes relative to TGFβ-induced collagen fibril synthesis, we employed our long-term (6 weeks) 3D in vitro model of lung fibrosis using primary human lung fibroblasts from patients with IPF, which we have previously described (*Jones et al., 2018*) and which allows direct evaluation of pyridinoline cross-linking, collagen nano-structure, and tissue biomechanics. We employed the selective HIF-prolyl hydroxylase inhibitor IOX2 to test within the in vitro fibrosis model, confirming HIF stabilisation by IOX2 following 2-week culture, and that in combination with TGFβ1 this promoted PLOD2 and LOXL2 expression (*Figure 5—figure supplement 1a* and b). Following 6 weeks of culture with TGFβ1 in the absence (control) or presence of IOX2 to drive HIF pathway activation, mature pyridinoline cross links (DPD/PYD) were significantly increased by the addition of IOX2 (*Figure 5a*) and these achieved a level comparable to our previous findings in IPF tissue (*Jones et al., 2018*). The biomechanical consequence of HIF stabilisation by IOX2 treatment was then investigated with parallel plate compression testing, identifying a greater than threefold increase in tissue stiffness by the addition of IOX2 (*Figure 5b*), with the mean (± SEM) compressive modulus measurement following IOX2 treatment of (107.1 ± 10.7) kPa comparable to the maximal stiffness of between 50 and 150 kPa we and others have previously identified in highly fibrotic areas in IPF tissue (*Booth et al., 2012*).

We next assessed collagen morphology. When visualised by polarised light Picrosirius red microscopy (*Figure 5c*), highly organised collagen fibrils were evident in vehicle-treated fibrotic control cultures as well as in those treated with IOX2 with no apparent morphological differences (*Figure 5—figure supplement 1c*). By contrast, ultrastructural analysis of the collagen fibrils using electron microscopy identified a change in collagen nanostructure with a significant decrease in fibril diameter (*Figure 5d and e*) when pyridinoline cross-linking was increased by IOX2, consistent with our previous observation that fibril diameter is increased by inhibition of pyridinoline cross-linking (*Jones et al., 2018*). In support of the disease relevance of our in vitro findings, non-hydrated collagen fibrils from patients with IPF have reduced diameters when measured by atomic force microscopy (*Figure 5f*), consistent with our previous findings that hydrated collagen fibrils extracted from IPF lung tissue have a reduced diameter compared to control samples (*Jones et al., 2018*). Together, these data identify HIF pathway activation to be a key regulator of pyridinoline cross-link density, collagen fibril nano-architecture, and tissue stiffness.

## Pseudohypoxia and loss of FIH activity promotes HIF pathway activation in lung fibroblasts

Whilst canonical HIF pathway activation was observed in lung fibroblasts under hypoxic conditions, elevated levels of HIF1α and HIF2α in IPF fibroblasts under normoxic conditions have recently been reported (*Aquino-Gálvez et al., 2019*), suggesting a pseudohypoxic state that is a state in which cells express, at least some, hypoxia-associated genes and proteins, regardless of the oxygen status (*Russell et al., 2017*). To further investigate this possibility, we employed gene set variation analysis (GSVA) using a validated 15-gene HIF/hypoxia gene expression signature (*Buffa et al., 2010*) to published datasets, identifying that fibroblasts cultured under normoxic conditions from patients with a usual interstitial pneumonia pattern of fibrosis or systemic sclerosis associated lung fibrosis have a

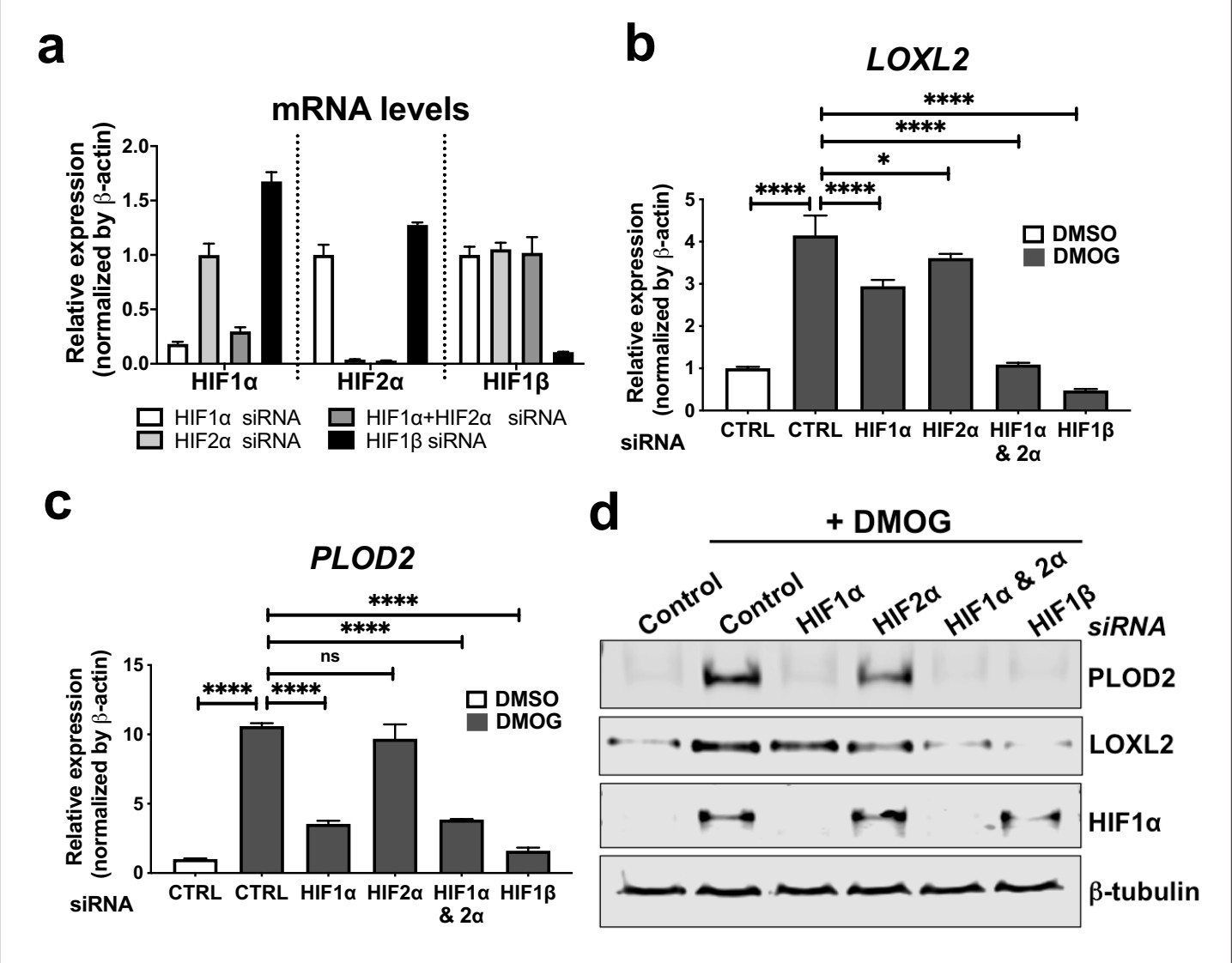

**Figure 3.** HIF pathway activation regulates PLOD2 and LOXL2 expression in lung fibroblasts from patients with IPF. (**A**) Fold changes in mRNA levels of HIF1α (*HIF1A*), HIF2α (*EPAS1*), and HIF1β (*ARNT*) in primary human lung fibroblasts from patients with IPF transfected with indicated siRNA followed by treatment with DMOG. β-actin-normalised mRNA levels in control cells were used to set the baseline value at unity. Data are mean ± s.d. n = 3 samples per group. (**B, C**) Fold change in mRNA levels of *LOXL2* (**B**) and *PLOD2* (**C**) in IPF fibroblasts transfected with indicated siRNA followed by treatment with DMOG or vehicle control. β-actin-normalised mRNA levels in control cells were used to set the baseline value at unity. Data are mean ± s.d. n = 3 samples per group. ns (not significant, $p > 0.05$); *$p < 0.05$; ****$p < 0.0001$ by Dunnett's multiple comparisons test. (**D**) PLOD2, LOXL2 and HIF1α and β-tubulin protein levels in IPF fibroblasts transfected with indicated siRNA followed by treatment of DMSO or DMOG. β-tubulin was used as a loading control. The full blots are shown in *Figure 3—source data 1*.

The online version of this article includes the following source data for figure 3:

**Source data 1.** Full membrane scans for western blot images for *Figure 3d*.

significantly increased HIF score (i.e. manifest evidence for HIF upregulation) compared to cultured control fibroblasts (*Figure 6a*), consistent with an oxygen independent increase in HIF activity. Furthermore, there was a significant increase in the HIF score in lung mesenchymal stromal cells of patients with progressive lung fibrosis compared to those with stable fibrosis (*Figure 6b*), suggesting that HIF pathway activation may be required for fibrosis progression.

To further investigate the mechanism underlying pseudohypoxic HIF activity in lung fibrosis, we investigated the role of Factor Inhibiting HIF (FIH), a Fe (II)- and 2-oxoglutarate (2-OG)-dependent dioxygenase, which regulates HIF activity and likely the set of HIF target genes upregulated via

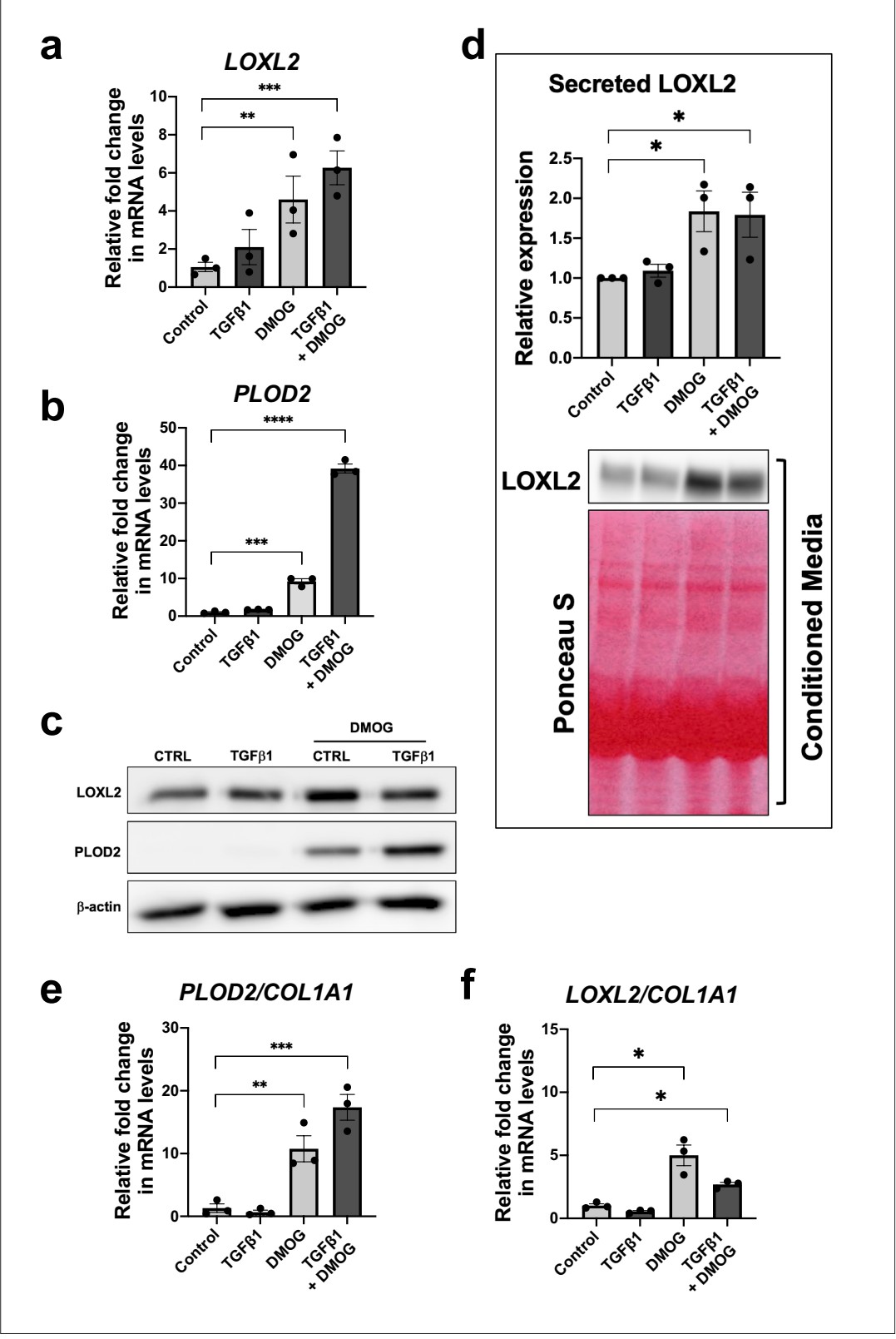

**Figure 4.** HIF pathway activation promotes *PLOD2* and *LOXL2* gene expression relative to fibrillar collagen expression. Lung fibroblasts from IPF donors (n = 3 across two independent experiments) were cultured in the presence or absence TGFβ1, DMOG, combined TGFβ1 and DMOG, or vehicle control for 48 hr. (**A, B**) Relative gene expression of *PLOD2* (**A**) and *LOXL2* (**B**) using the ΔΔCt method. Bars indicate geometric means. Data are

*Figure 4 continued on next page*

*Figure 4 continued*

mean ± s.d. **p < 0.01; ***p < 0.001; ****p < 0.0001 by Dunnett's multiple comparisons test. (**C**) PLOD2 and LOXL2 protein levels. β-actin was used as a loading control. (**D**) Protein expression of LOXL2 in conditioned media. Ponceau S staining showing total protein levels. The full blots are shown in *Figure 4—source data 1*. Bars in graph indicate geometric means. Data are mean ± s.d. **p < 0.01; ***p < 0.001; ****p < 0.0001 by Dunnett's multiple comparisons test. (**E, F**) Expression of *PLOD2* and *LOXL2* from (**A and B**) was divided by *COL1A1* expression (shown in *Figure 4—figure supplement 2*) to calculate proportionate expression changes of cross-linking enzymes relative to collagen fibrillogenesis gene expression. Bars indicate geometric mean. Grouped analysis was performed using Dunnett's multiple comparison test. * p < 0.05, ** p < 0.01, *** p < 0.001, ****p < 0.0001.

The online version of this article includes the following source data and figure supplement(s) for figure 4:

**Source data 1.** Full membrane scans for western blot images for *Figure 4a and b*.

**Figure supplement 1.** HIF stabilisation increases LOXL2 secretion in control fibroblasts.

**Figure supplement 1—source data 1.** Full membrane scans for western blot images for *Figure 4—figure supplement 1*.

**Figure supplement 2.** TGFβ1 promotes interstitial collagen gene expression in lung fibroblasts.

hydroxylating a conserved asparagine (Asn) residue within the HIFα C-terminal activation domain (CAD), a post-translational modification that blocks interactions between the HIFα-CAD and the histone acetyl transferases CBP/p300 (*Elkins et al., 2003*; *Hewitson et al., 2002*; *Lando et al., 2002*; *Mahon et al., 2001*; *McNeill et al., 2002*; *Chan et al., 2016*). Whilst oxygen tension is the classical regulator of FIH activity, oxidative stress can also inactivate FIH so promoting HIF activity under normoxic conditions (*Masson et al., 2012*).

Initially, to investigate the potential role of reduced FIH activity in regulating collagen post-translational modifications, we silenced FIH under normoxic conditions; the results show that loss of FIH was sufficient to induce both PLOD2 and LOXL2 expression, and that this effect required HIF promoted transcription, since HIF1β knockdown prevented their induction (*Figure 6c*). Whilst FIH is stably constitutively expressed across tissues (*Bracken et al., 2006*; *Stolze et al., 2004*), the activity levels of FIH can vary (*Wang et al., 2018*; *Tan et al., 2007*; *Kroeze et al., 2010*); thus, we compared FIH activity in control or IPF fibroblasts using a UAS-luc/GAL4DBD-HIF1αCAD binary reporter system (HIF1α CAD reporter) (*Coleman et al., 2007*). In this assay, the activity of FIH is monitored by a Gal4-driven luciferase reporter that registers the activity of the heterologous Gal4-HIF-CAD fusion protein. Inhibition of FIH leads to a reduction in hydroxylation at Asn-803 of the HIF-CAD (C-terminal trans-activation domain) fusion, which permits increased recruitment of the transcriptional co-activators p300/CBP and enhanced reporter gene activity (*Figure 6d*). Consistent with a loss of function of FIH in lung fibrosis, we found FIH activity was significantly reduced in fibroblasts from patients with IPF compared to control fibroblasts (*Figure 6e*). We further confirmed that a reduction in FIH activity in normal lung fibroblasts could be caused under normoxia by oxidative stress, achieving a level of HIF CAD activity comparable to treatment with the hypoxia mimetic DMOG (*Figure 6f*). Thus, in lung fibroblasts a reduction in FIH activity may promote HIF pathway activation to dysregulate collagen structure-function.

We next employed the FIH-selective inhibitor DM-NOFD (*McDonough et al., 2005*) within our 3D model of fibrosis. We confirmed that FIH inhibition by DM-NOFD was sufficient to induce the HIF pathway activation marker gene carbonic anhydrase IX (*CA9*), *PLOD2*, and *LOXL2* gene expression following 2-week culture (*Figure 6—figure supplement 1*), and in combination with HIF stabilisation (IOX2) this expression was further increased. Following 6 weeks of culture, DM-NOFD increased mature pyridinoline cross-links (*Figure 6g*) as well as tissue stiffness (*Figure 6h*), whilst the combination of DM-NOFD and IOX2 was additive. Thus, FIH inhibition can promote collagen post-translational modification and increase tissue stiffness.

## HIF pathway activation localises in areas of active fibrogenesis to cells co-expressing LOXL2 and PLOD2

To support our in vitro studies, we investigated for evidence that HIF regulates *PLOD2* and *LOXL2* expression within the fibroblast foci of human IPF lung tissue. To assess for HIF activity, we applied GSVA using the 15-gene HIF/hypoxia gene expression signature (*Buffa et al., 2010*) to the transcriptome

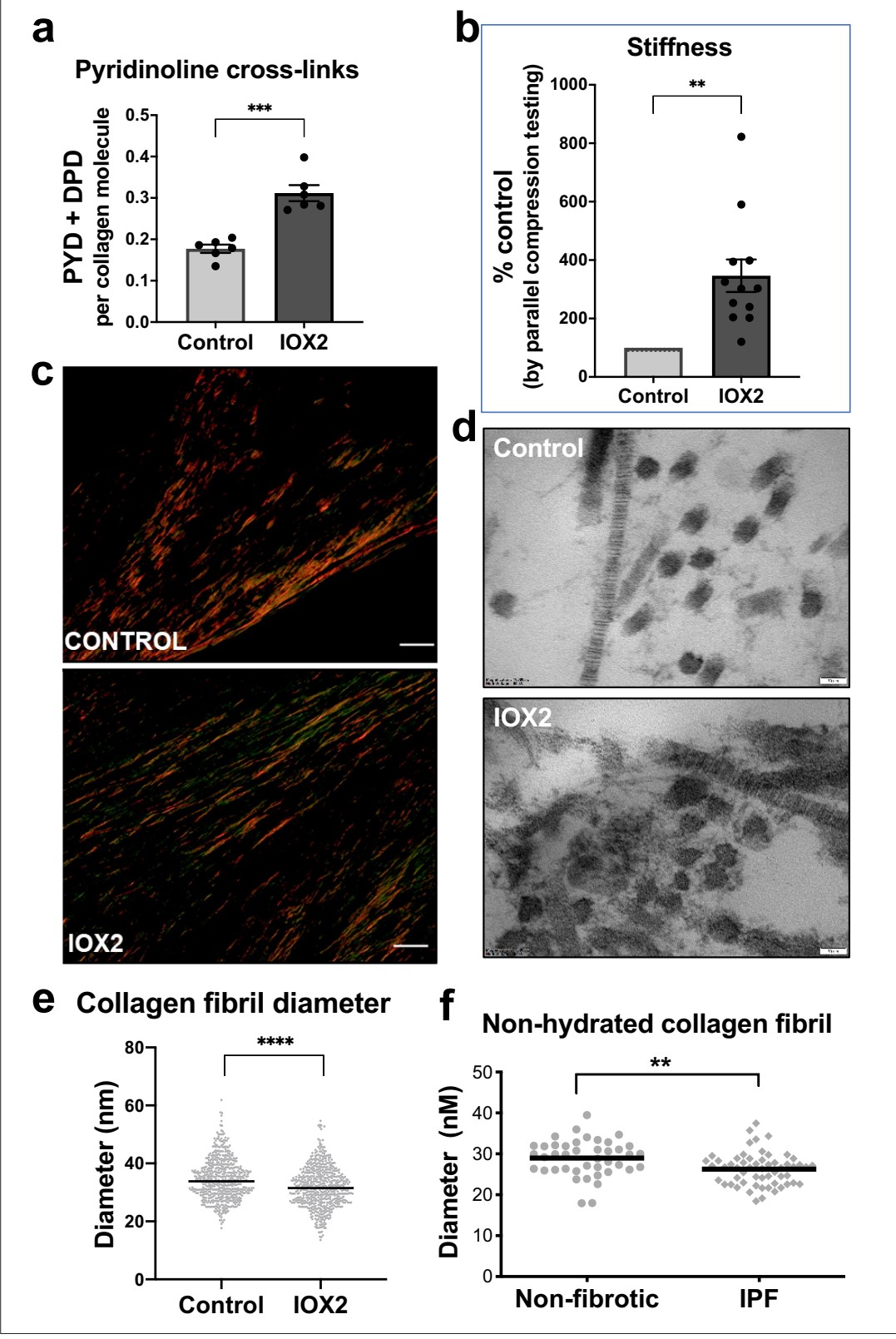

**Figure 5.** HIF pathway activation promotes pyridinoline cross-linking, alters collagen nano-architecture, and increases tissue stiffness. Lung fibroblasts from IPF patients (n = 3 donors, two experiments per donor) were used in the 3D model of fibrosis in the presence of IOX2 or vehicle control. Bars indicate geometric mean + s.e.m. Analysis was performed using a Mann-Whitney t-test (two-tailed) **p < 0.01; ***p < 0.001; ****p < 0.0001.

*Figure 5 continued on next page*

*Figure 5 continued*

(**A**) Total mature trivalent (PYD+ DPD) collagen cross-links determined by ELISA. n = 6 samples from three IPF donors. (**B**) Tissue stiffness measured from parallel-plate compression testing (n = 12 samples from three IPF donors) determined by Young's modulus and represented as proportion of control. (**C**) Representative images of histological sections of samples stained with picrosirius red and imaged under polarised light. Scale bar 20 μm. (**D**) Representative electron microscopy images of collagen fibrils within the 3D model of fibrosis. Scale bar 50 nm. (**E**) Collagen fibril diameter within the 3D model of fibrosis measured in transverse section (300 fibrils for each condition from two IPF donors, measured by a blinded investigator). (**F**) Atomic force microscopy indentation modulus of collagen fibrils (3–7 fibrils per donor) from control (n = 42 fibrils from eight donors) or IPF lung tissue (n = 57 fibrils from 10 donors) under non-hydrated conditions; each data point represents the mean of 30–50 force-displacement curves per fibril.

The online version of this article includes the following source data and figure supplement(s) for figure 5:

**Figure supplement 1.** IOX2-mediated HIF pathway activation promotes PLOD2 and LOXL2 expression in the 3D in vitro model of fibrosis.

**Figure supplement 1—source data 1.** Full membrane scans for western blot images for *Figure 5—figure supplement 1a, b*.

of each fibroblast focus, identifying that the HIF signature score, but not TGFβ score, significantly correlated with *LOXL2/PLOD2* expression (*Figure 7a and b*). Furthermore, analysis of serial tissue sections using immunohistochemistry identified that HIF1α and the HIF pathway activation marker gene carbonic anhydrase IX (CA-IX) were expressed within fibroblast foci (*Bodempudi et al., 2014*; *Loncaster et al., 2001*), and that this expression localised to cells co-expressing *LOXL2* and *PLOD2* mRNA (*Figure 7c*; *Figure 7—figure supplement 1*). Finally, as FIH is more sensitive to inhibition by oxidative stress (*Masson et al., 2012*) compared to the PHDs, which are more sensitive to hypoxia than FIH (*Masson et al., 2012*), we investigated whether HIF activation occurs in lung mesenchymal cells in the context of oxidative stress. We applied GSVA to a published single cell RNAseq dataset (114,396 cells) from 10 control and 20 fibrotic lungs which identified 31 cell types including four fibroblast types (fibroblasts, myofibroblasts, Hyaluronan Synthase 1 (HAS1) high and Perilipin 2 (PLIN2)+ fibroblasts) (*Figure 8—figure supplement 1*; *Habermann et al., 2020*). Applying HIF signature or upregulated oxidative stress gene expression signatures to this dataset, we identified that, compared to fibroblasts and myofibroblasts, the HAS1 high and PLIN2+ cells, whose presence was almost exclusively derived from the IPF lung tissue, had significantly increased HIF and upregulated oxidative stress scores (*Figure 8a–d*) and that these two scores were significantly correlated (*Figure 8e*), consistent with an increase in pseudohypoxic HIF activity in these disease-specific mesenchymal cell types.

## Discussion

We previously reported that altered collagen fibril nanoarchitecture is a core determinant of dysregulated ECM structure-function in human lung fibrosis (*Jones et al., 2018*). Here, through ex vivo models, bioinformatics and human lung fibrosis tissue studies, we extend these observations leading to the discovery that HIF pathway activation promotes pathologic pyridinlone collagen crosslinking and tissue stiffness by disproportionate induction of collagen-modifying enzymes relative to TGFβ-induced collagen fibril synthesis. Furthermore, this may occur via pseudohypoxic oxygen-independent mechanisms, including the involvement of a decrease in FIH activity that can occur due to oxidative stress, which is thought to play a significant role in IPF pathogenesis (*Cheresh et al., 2013*). Consistent with this, oxidative stress is increased in subpopulations of IPF fibroblasts whilst FIH activity is significantly reduced in fibroblasts from patients with lung fibrosis resulting in HIF activation under normoxic conditions. Thus, we provide evidence that dysregulated HIF activity is a core regulator of ECM structure-function in human lung fibrosis, and that this may be a key determinant of pathologic tissue stiffness and progressive human lung fibrosis.

TGFβ is a multifunctional growth factor with key roles in normal development and wound healing. It is also considered the prototypic profibrogenic cytokine that promotes increased ECM deposition and has been associated with fibrosis across multiple organs (*Yue et al., 2010*). We identified that in lung fibroblasts, TGFβ1 increased fibrillar collagen mRNA transcription but its relative effects on *PLOD2* or *LOXL2* were more limited, suggesting that TGFβ pathway activation alone may be insufficient

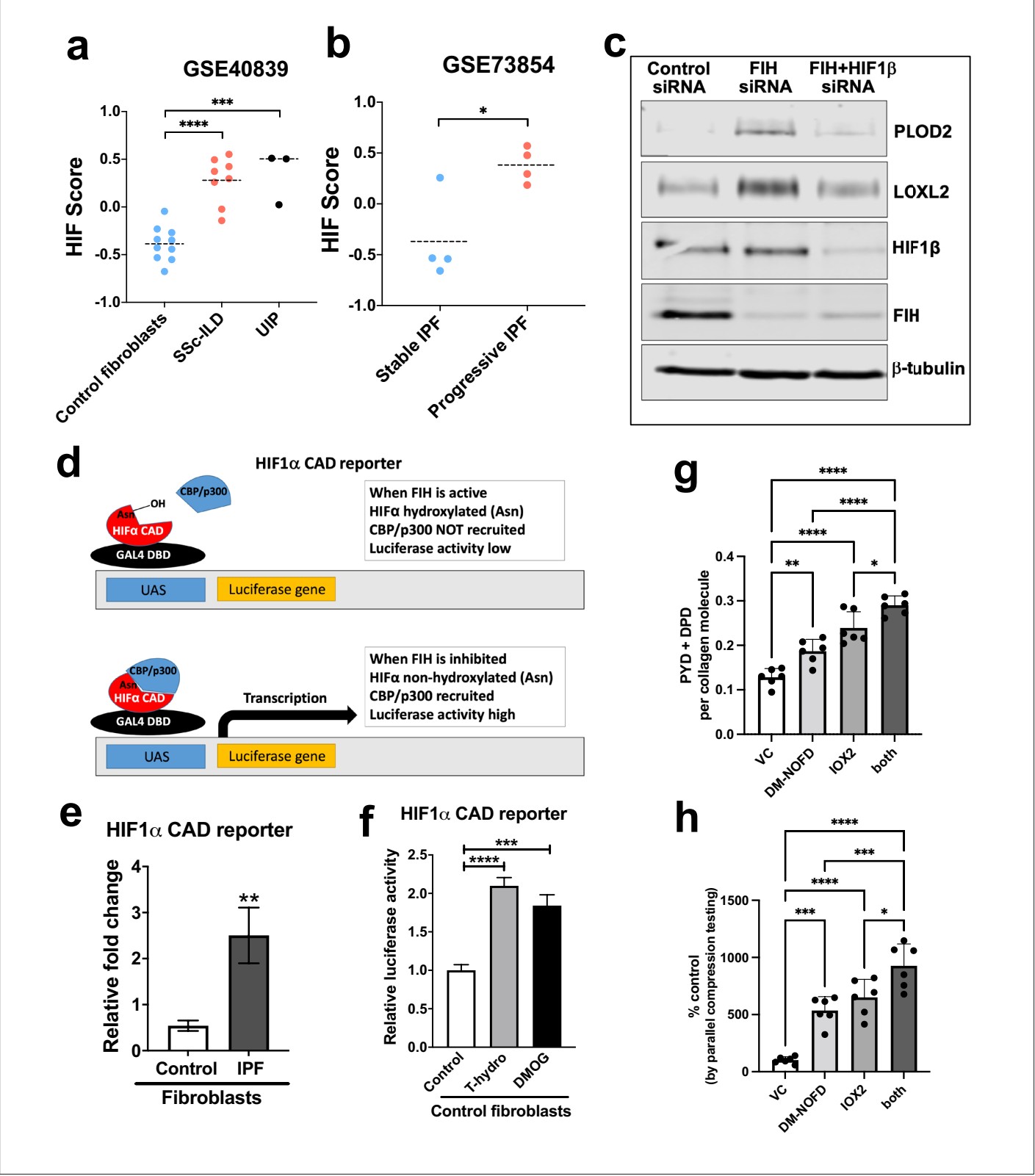

**Figure 6.** Pseudohypoxia and loss of FIH activity promotes HIF pathway signalling in IPF fibroblasts and increases tissue stiffness. (**A**) HIF GSVA scores calculated in human lung fibroblasts derived from control or patients with interstitial lung disease (scleroderma lung or a usual interstitial pneumonia / IPF pattern) (GSE40839). Data are mean ± s.d. ***p < 0.001; ****p < 0.0001 by Dunnett's multiple comparisons test. (**B**) HIF GSVA scores calculated in human bronchoalveolar lavage derived mesenchymal stromal cells from patients with stable and progressive IPF (GSE73854). Data are mean ± s.d.

*Figure 6 continued on next page*

*Figure 6 continued*

*p < 0.05 by the unpaired t test. (**C**) PLOD2, LOXL2, HIF1β, FIH, and β-tubulin protein levels in lung fibroblasts from patients with IPF transfected with indicated siRNA. β-tubulin was used as a loading control. The full blots are shown in *Figure 6—source data 1*. (**D**) Diagram explaining the HIF1α CAD reporter assay in **E** and **F**. In brief, the FIH asparaginyl hydroxylase hydroxylates HIF1α CAD, inhibiting its binding with CBP/p300 and decreasing luciferase activity. When FIH is inhibited, the non-hydroxylated HIF1α CAD can bind with CBP/p300 increasing luciferase activity. (**E**) HIF1α CAD reporter assays in normal human lung fibroblasts (control fibroblasts) or IPF lung fibroblasts (IPF fibroblasts). Values represent the relative fold increase of firefly luciferase in relation to Renilla luciferase, normalised against control (1.0). Data are mean ± s.d. n = 3 samples per group. **p < 0.01 by unpaired t test. (**F**) HIF1α CAD reporter assays in control fibroblasts with indicated treatment (hydrogen peroxide (T-hydro), DMOG, or vehicle control). Values represent relative fold of firefly luciferase in relation to *Renilla* luciferase, normalised against control (1.0). Data are mean ± s.d. n = 3 samples per group. (**G and H**) Control lung fibroblasts (n = 3 donors, two experiments per donor) were used in the 3D model of fibrosis in the presence of IOX2 and/ or DM-NOFD or vehicle control as indicated. (**G**) Total mature trivalent (PYD+ DPD) collagen cross-links determined by ELISA. n = 6 samples from three donors. (**H**) Tissue stiffness measured from parallel-plate compression testing (n = 6 samples from three donors) determined by Young's modulus and represented as proportion of control. * p < 0.05, ** p < 0.01, *** p < 0.001, ****p < 0.0001 by Dunnett's multiple comparisons test.

The online version of this article includes the following source data and figure supplement(s) for figure 6:

**Source data 1.** Full membrane scans for western blot images for *Figure 6*.

**Figure supplement 1.** Pseudohypoxia and loss of FIH activity promotes HIF pathway signalling and increases LOXL2 and PLOD2 expression.

to cause a substantial increase in pathologic pyridinoline collagen crosslinking. HIF-mediated transcription appears to be relatively more important in inducing PLOD2/LOXL2 relative to interstitial collagen fibril synthesis, so promoting pyridinoline collagen cross-linking, altering collagen fibril nanostructure, and increasing tissue stiffness. While TGFβ has been reported to cause HIF stabilisation (*Basu et al., 2011*), our findings suggest that this effect is modest and that further HIF-mediated activation is likely required to drive matrix stiffening. This proposal is consistent with a recent result implying a hierarchical relationship in which HIF proteins play a relatively important role in the induction of PLOD2 expression, that is in this regard the effect of the HIF transcription factors appears to be more important relative to that of TGFβ stimulated SMAD proteins (*Rosell-García et al., 2019*). Thus, we propose that HIF pathway activation acts as a key pathologic 'second hit' which disrupts the normal wound healing role of TGFβ by altering collagen fibril nanoarchitecture so dysregulating ECM structure-function and promoting progressive lung fibrosis. In keeping with this concept, GSVA using a validated HIF score (*Buffa et al., 2010*) applied to microarray data for lung mesenchymal stromal cells showed that HIF activity was increased in cells from patients with progressive lung fibrosis compared with those with stable disease.

We investigated the functional consequences of our findings by employing our long-term 3D in vitro model of lung fibrosis. The results show that HIF pathway activation using a HIF stabilising PHD inhibitor and/or an FIH inhibitor increased pyridinoline cross-links to a level comparable to that identified in IPF tissue, and that the increase in cross-links is associated with an increase in tissue stiffness comparable to the extremes of stiffness identified in IPF tissue together with a reduction in fibril diameter similar to those present in IPF lung tissue. Together these observations support the human disease relevance of HIF pathway activation to IPF and define conditions for future mechanistic studies whereby the 3D in vitro model recapitulates key features of dysregulated collagen structure-function in IPF.

The LOX and LOXL enzymes play key roles in the process of fibrillar collagen production and are tightly regulated in normal development and under physiological conditions (*Trackman, 2016*). In our LCMD RNA-Seq analyses, LOXL2 was the most highly expressed LOX/LOXL family member as well as the only LOX/LOXL member which correlated with PLOD2 expression, whilst in our previous work investigating collagen structure-function dysregulation in human lung fibrosis, we identified that gene expression of LOXL2 was significantly increased in IPF tissue when compared to age-matched control lung tissue (*Jones et al., 2018*). Furthermore, using a small molecule LOXL inhibitor in our 3D model of fibrosis, we identified a greater than 50% reduction in mature pyridinoline cross-links using a concentration which completely inhibits LOXL2 but has minimal effects on LOX and LOXL1 (*Jones et al., 2018*). This is consistent with previous reports that LOXL2 has key pathologic roles in cancer and fibrosis (*Barker et al., 2012*; *Barry-Hamilton et al., 2010*). As our studies do not unequivocally exclude a potential role for other LOX/LOXL family members in human lung fibrosis an area of future study could be the systematic silencing of each LOX/LOXL family member using CRISPR gene editing.

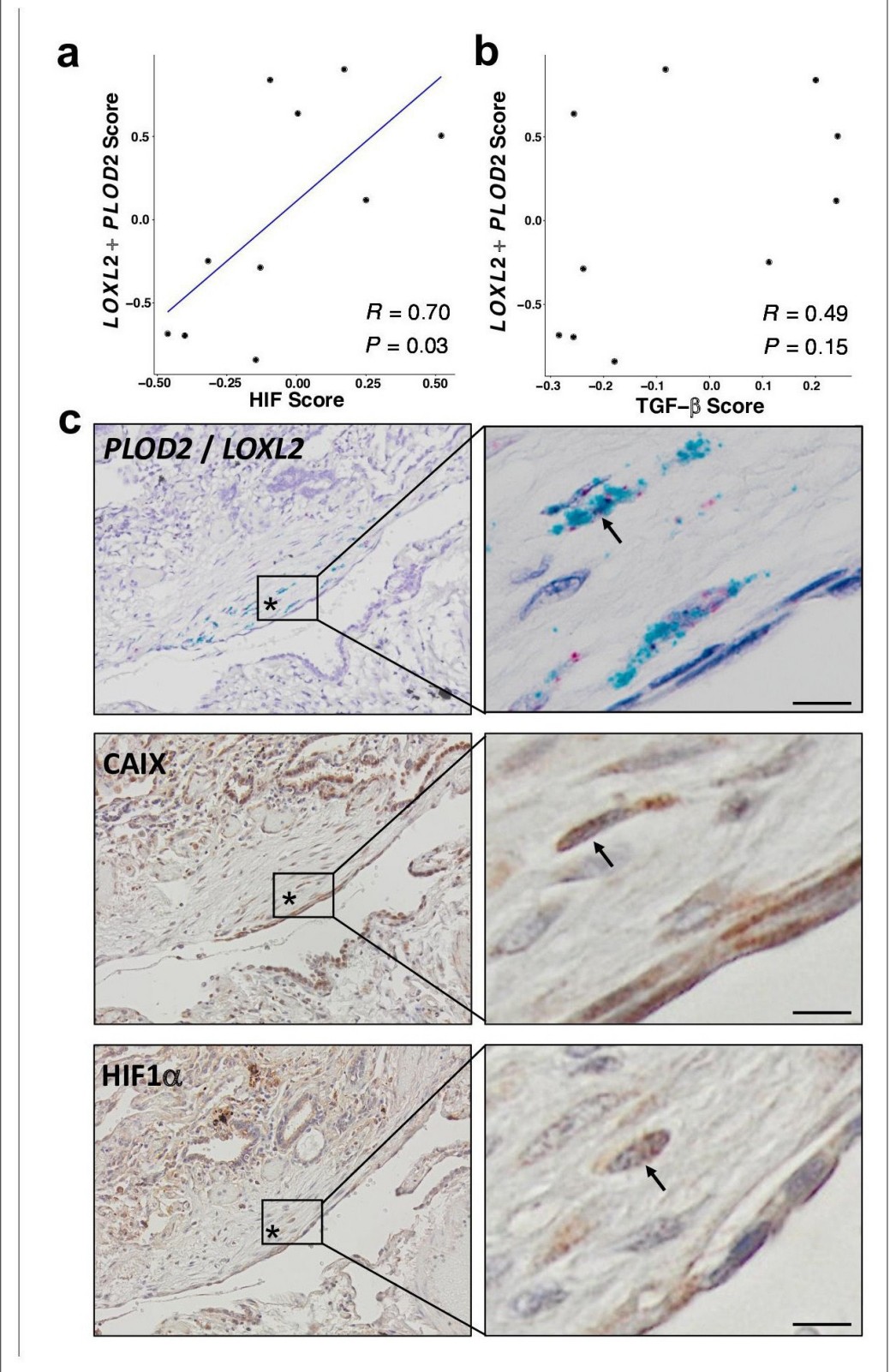

**Figure 7.** HIF pathway activation localises in areas of active fibrogenesis to cells co-expressing LOXL2 and PLOD2. (**A–B**) Scatterplots showing correlations between *LOXL2/PLOD2* expression and HIF scores (**A**) or TGFβ scores (**B**) in IPF fibroblast foci (n = 10) using the Spearman rank correlation coefficient. (**C**) Representative images of serial sections of lung tissue from patients with IPF (n = 3). mRNA expression of *PLOD2* (red chromagen) and *LOXL2*

*Figure 7 continued on next page*

*Figure 7 continued*

(green chromagen) using RNAscope RNA in-situ hybridisation with immunohistochemical staining for Carbonic anhydrase IX (CA-IX) and HIF1α using DAB (brown). A fibroblastic focus is identified by *. Scale bar 20 µm.

The online version of this article includes the following figure supplement(s) for figure 7:

**Figure supplement 1.** HIF pathway activation localises in areas of active fibrogenesis to cells co-expressing LOXL2 and PLOD2.

The HIF signalling pathway has been reported to be active in lungs and fibroblasts from IPF patients, as determined by the abundance of HIF1α and HIF2α (*Aquino-Gálvez et al., 2019*; *Bodempudi et al., 2014*). These findings are consistent with our own observations of increased expression of the HIF-responsive gene, CA-IX. Hypoxia has been proposed to have a pathogenetic role in lung fibrosis through mechanisms including fibroblast proliferation, augmented ER stress, epithelial-mesenchymal transition, and glycolytic reprogramming (*Bodempudi et al., 2014*; *Higgins et al., 2007*; *Senavirathna et al., 2018*; *Goodwin et al., 2018*). Furthermore, a number of reports have proposed that cross-talk between TGFβ and hypoxia may promote fibrosis, with hypoxia and TGFβ1 synergistically increasing myofibroblast marker expression (*Senavirathna et al., 2020*), promoting experimental nickel oxide nanoparticle-induced lung fibrosis (*Qian et al., 2015*), and HIF1α mediating TGF-β-induced PAI-1 production in alveolar macrophages in the bleomycin model of lung fibrosis (*Ueno et al., 2011*). Here, we extended these previous observations by showing that in lung fibrosis, loss of FIH activity either by siRNA-mediated knockdown or exposure to oxidative stress induces HIF pathway activation independently of oxygen tension, so dysregulating collagen fibrillogenesis under normoxic conditions. FIH negatively regulates HIF activity by hydroxylation of N803, preventing the interaction of the HIFα CAD with CBP/p300 (*Elkins et al., 2003*; *Hewitson et al., 2002*; *Lando et al., 2002*; *Mahon et al., 2001*; *McNeill et al., 2002*). Whilst oxygen tension is the classical regulator of FIH activity, oxidative stress can inactivate FIH so promoting HIF activity, with FIH more sensitive to oxidative stress than the HIF prolyl hydroxylases (*Masson et al., 2012*). Oxidative stress has been implicated as an important profibrotic mechanism in the lungs and other organs (*Cheresh et al., 2013*; *Purnomo et al., 2013*; *Sánchez-Valle et al., 2012*); it can arise from exposure to environmental toxins (e.g. air pollution, tobacco, asbestos, silica, radiation, and drugs such as bleomycin) or from endogenous sources including mitochondria, NADPH oxidase (NOX) activity, and/or inadequate or deficient antioxidant defenses (*Cheresh et al., 2013*). In our bioinformatic studies, we observed subsets of disease-specific fibroblasts with elevated scores for oxidative stress and these same populations had evidence of HIF pathway activation. Further investigation is merited to understand the consequences of this upon the fibrotic microenvironment including possible dysregulation of epithelial-mesenchymal cross-talk.

To our knowledge whether perturbations in FIH activity could contribute to fibrosis has not been investigated previously. Whilst our studies have focused upon HIF pathways and collagen, functionally FIH, via both HIF-dependent and HIF-independent pathways, has been reported to regulate metabolism (*Zhang et al., 2010*; *Scholz et al., 2016*; *Peng et al., 2012a*; *Sim et al., 2018*), keratinocyte differentiation (*Peng et al., 2012b*), vascular endothelial cell survival (*Kiriakidis et al., 2015*), tumour growth (*Pelletier et al., 2012*; *Kuzmanov et al., 2012*) and metastasis (*Kang et al., 2018*) as well as Wnt signalling (*Rodriguez et al., 2016*), suggesting that the loss of FIH activity that we have identified could have pleiotropic effects in lung fibrosis, meriting further investigation.

In summary, this study identifies that HIF pathway activation via oxygen dependent and oxygen independent mechanisms promotes pyridinoline collagen cross-linking which is a defining feature of human lung fibrosis that dysregulates ECM structure-function to promote progressive lung fibrosis. Our findings suggest that therapeutically targeting HIF pathway activation might restore ECM homeostasis and so prevent fibrosis progression.

## Materials and methods
### Lung tissue sampling
Human lung experiments were approved by the Southampton and South West Hampshire and the Mid and South Buckinghamshire Local Research Ethics Committees (ref 07 /H0607/73), and all subjects

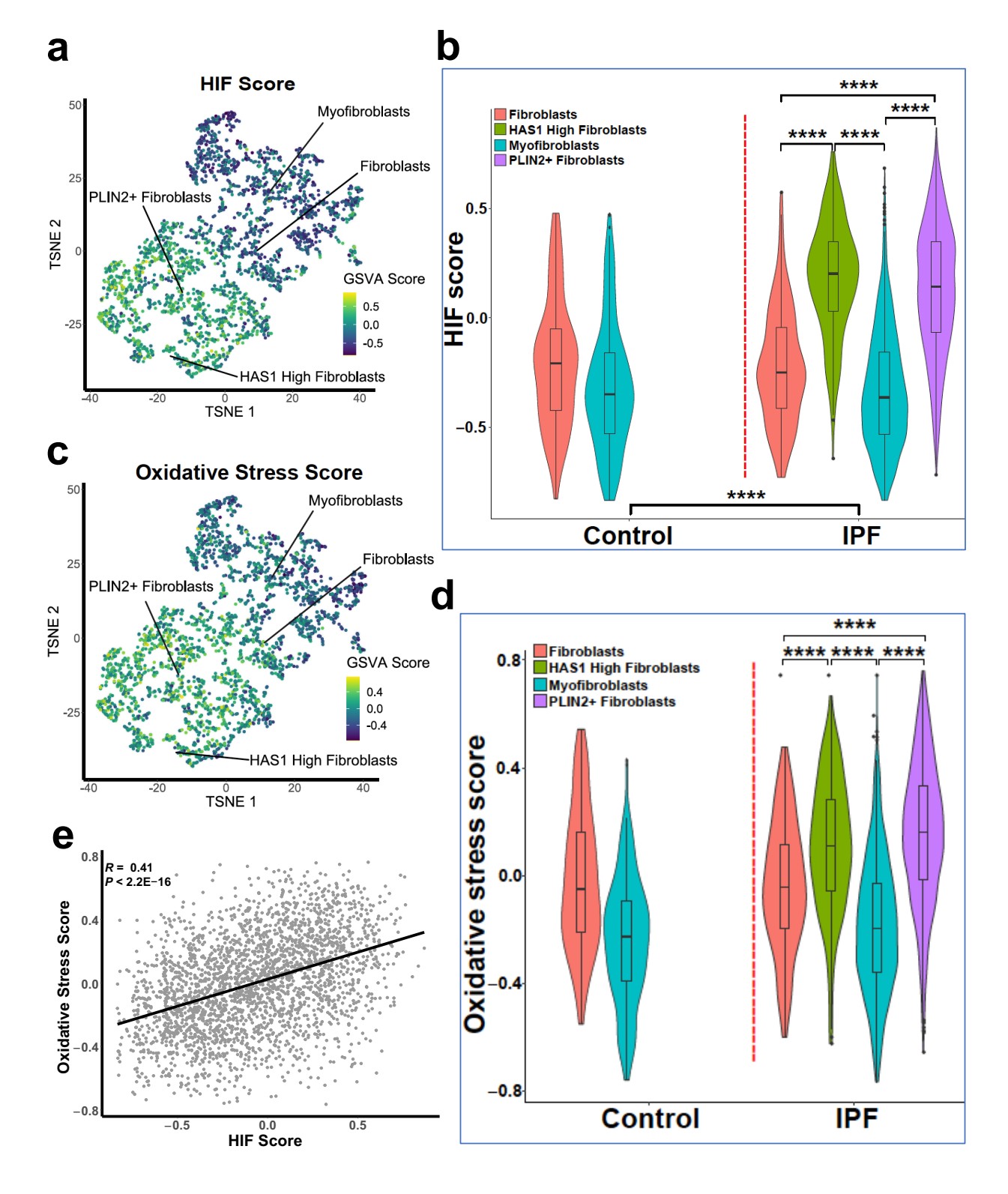

**Figure 8.** Gene set variance analysis of single-cell RNAseq fibroblast populations identifies co-enrichment of HIF score and oxidative stress genes. (**A**) HIF score GSVA in control and IPF fibroblasts sequenced by single-cell RNAseq (GSE135893). Colours correspond to calculated GSVA score for each cell. (**B**) Plot of mean HIF GSVA scores for each fibroblast type in control and IPF fibroblast cell populations and compared using Dunnett's multiple comparison test, ****p < 0.0001. (**C**) GSVA scores for genes upregulated in IPF in this dataset associated with the Gene Set: HALLMARK_

*Figure 8 continued on next page*

*Figure 8 continued*

REACTIVE_OXYGEN_SPECIES_PATHWAY (M5938). (**D**) Plot of upregulated oxidative stress GSVA scores for each fibroblast type in control and IPF cells. (**E**) Correlation plot of HIF score vs upregulated oxidative stress GSVA score for single cell RNAseq data. Correlation coefficient is Pearson's product-moment coefficient.

The online version of this article includes the following figure supplement(s) for figure 8:

**Figure supplement 1.** Fibroblast populations identified within a single-cell RNA sequencing dataset.

gave written informed consent. Clinically indicated IPF lung biopsy tissue samples deemed surplus to clinical diagnostic requirements were formalin fixed and paraffin embedded. All IPF samples were from patients subsequently receiving a multidisciplinary diagnosis of IPF according to international consensus guidelines.

## Transcriptomic analysis of in situ IPF fibroblast foci

We analysed a transcriptomic data set that we have recently established (GSE169500). Briefly, laser capture microdissection was performed upon Formalin-Fixed Paraffin-Embedded (FFPE) control non-fibrotic lung tissue (alveolar septae, [n = 10]) and usual interstitial pneumonia/idiopathic pulmonary fibrosis FFPE lung tissue (fibroblast foci, [n = 10] and adjacent non-affected alveolar septae, [n = 10]). Total RNA was isolated, cDNA libraries were prepared using Ion Ampli-Seq-transcriptome human gene expression kit (Life Technologies, Paisley, UK) and sequenced using Ion Torrent Proton Sequencer. A two-stage mapping strategy was used to map the reads to UCSC hg19 human genome. Cufflinks was used to calculate Fragments per Kilobase of exon per Million (FPKM) values.

## RNA in-situ hybridisation

Simultaneous in situ detection of the *LOXL2* and *PLOD2* mRNA on human IPF formalin-fixed paraffin-embedded tissue sections from IPF lung tissue biopsy samples from seven patients were performed using duplex RNAscope technology (Advanced Cell Diagnostics, Biotechne, Abingdon, UK). *LOXL2* was detected by C1-probe (Probe-Hs-LOXL2-C1, 311341) and *PLOD2* was detected by C2-probe (Probe-Hs-PLOD2-C2, 547761-C2). Briefly, 5 µm human IPF lung tissue sections were baked at 60 °C, deparaffinised in xylene, followed by dehydration in an ethanol series. Target retrieval, hybridisation with target probes, amplification, and chromogenic detection were performed according to the manufacturer's recommendations (RNAscope 2.5 Duplex Detection protocol for FFPE tissues). Sections were counterstained with Gill's Hematoxylin, and mounted with Vectamount permanent mounting medium prior to imaging. Assays were performed with duplex positive (*PPIB* and *POLR2A*) and negative controls. For co-localisation studies, adjacent serial sections were stained using modified Movat's pentachrome or hematoxylin and eosin stain as previously reported (*Jones et al., 2016*). Images were acquired using an Olympus Dotslide Scanner VS110 (Olympus UK, Southend-on-Sea, UK). Semi-quantitative analysis (range absent to +++) of LOXL2 and PLOD2 expression in cell types in IPF tissue was performed by an expert lung pathologist (AF).

## 2D cell culture, reagents, and transfections

Primary fibroblast cultures were established from lung parenchyma tissue of patients with IPF obtained by video-assisted thoracoscopic lung biopsy at University Hospital Southampton or non-fibrotic control lung parenchyma tissue (macroscopically normal lung sampled remote from a cancer site in patients undergoing surgery for early stage lung cancer) (*Jones et al., 2018*; *Yao et al., 2019*; *Conforti et al., 2020*; *Hill et al., 2019b*). MRC5 lung fibroblasts (RRID:CVCL_0440) were obtained from the European Collection of Authenticated Cell Cultures (ECACC). All cultures were tested and free of mycoplasma contamination. Demographic details for the primary lung fibroblast lines are provided in *Supplementary file 1b*.

Fibroblasts were cultured in Dulbecco's Modified Eagle's Medium (DMEM) supplemented with 10% foetal bovine serum (FBS), 50 units/ml penicillin, 50 µg/ml streptomycin, 2 mM L-glutamine, 1 mM sodium pyruvate, and 1 x non-essential amino acids (DMEM/FBS) (Life Technologies, Paisley, UK). All cells were kept at 37 °C and 5% $CO_2$. Hypoxic incubation of cells was carried out in a H35 Hypoxystation (Don Whitley Scientific) in which cells were cultured in humidified atmosphere of 1%

$O_2$, 5% $CO_2$, and 94% N2 at 37 °C. Following hypoxic incubation, cells were kept in hypoxic condition until samples were collected.

For pro-fibrogenic mediator studies, control lung fibroblasts were treated in the presence of EGF (R&D systems, 236-GMP-200, 10 ng/mL), TGFβ1 (R&D systems, 240-GMP-010, 10 ng/mL), Dimethyloxaloylglycine (DMOG) (Merck, CAS89464-63-1, 1 mM), Wnt3a (R&D systems, 5036-WN-010, 100 ng/mL), Wnt5a (R&D systems, 645-WN-010, 100 ng/mL), or vehicle control (DMSO). For subsequent HIF studies fibroblasts were treated in the presence of DMOG (1 mM), IOX2 (50 μM or 250 μM), or vehicle control (DMSO).

Short interfering RNA (siRNA) oligos against HIF1A (HIF1α) (MU-00401805-01-0002), EPAS1 (HIF2α) (MU-004814-01-0002), ARNT (HIF1β) (MU-007207-01-0002) and HIF1AN (FIH) (MU-004073-02-0002), LOXL2(L-008020-01-0005) were from Dharmacon, Cambridge, UK. Sequences are available from Dharmacon, or *Supplementary file 2*. As a negative control, we used siGENOME RISC-Free siRNA (Dharmacon, D-001220–01). Human lung fibroblasts were transfected with the indicated siRNA at a final concentration of 35 nM using Lipofectamine RNAiMAX reagent (Invitrogen).

## Reporter assay

FIH activity was evaluated using a UAS-luc/GAL4DBD-HIF1αCAD binary reporter system (HIF1α CAD reporter) (*Coleman et al., 2007*). For the luciferase reporter assays, human lung fibroblasts (control or IPF fibroblasts) were reverse transfected using Lipofectamine 3000 (Invitrogen) with 50 ng of phRL-CMV (Promega UK, Southampton, UK), which constitutively expresses the *Renilla* luciferase reporter, plus 225 ng of plasmid-GAL4DBD-HIF1αCAD and 225 ng of plasmid-UAS-luc per well. After 24 hour of transfection, a final concentration of 1 mM of DMOG, 1 mM DMSO or 20 μM freshly prepared T-hydro (tert-butyl hydroperoxide) (Sigma-Aldrich, Poole, UK) was dosed for 16 hours. T-hydro was added to the cells every 2 hours. Finally, the transcriptional assay was carried out using the Dual-Luciferase reporter assay system (Promega) following the manufacturer's protocol.

## HIF score, TGFβ score, and oxidative stress GSVA analyses

Raw CEL files for GSE73854 and GSE40839 were downloaded from GEO and imported into RStudio (version 3.6). Raw data were normalised by Robust Multi-array Average (RMA) function in the affy package (version 1.64.0). Multiple probes relating to the same gene were deleted and summarised as the median value for further analysis.

A 15-gene expression signature (*ACOT7, ADM, ALDOA, CDKN3, ENO1, LDHA, MIF, MRPS17, NDRG1, P4HA1, PGAM1, SLC2A1, TPI1, TUBB6*, and *VEGFA*) was selected to classify HIF activity (*Buffa et al., 2010*). All parameters and variables can be found in the accompanying file (*Source code 1*). This gene signature was defined based on knowledge of gene function and analysis of in vivo co-expression patterns and was highly enriched for HIF-regulated pathways. The HIF score for each sample was calculated by using gene set variation analysis (GSVA) (*Hänzelmann et al., 2013*) based on this 15-gene expression signature. The TGFβ score for each sample was calculated by using GSVA based on a list of gene from Gene Set: HALLMARK_TGF_BETA_SIGNALING (M5896). All parameters and variables can be found in the accompanying file (*Source code 2*). The Student t-test was used to evaluate the statistical difference in HIF scores between different conditions.

For single-cell transcriptomic analyses raw CEL files for GSE135893 were downloaded from GEO. Data was processed using the Seurat R package (v3.2.1) in R version 4.0.2. Cell types were assigned based on the published metadata (*Habermann et al., 2020*). Fibroblast counts data were log-normalised, variable genes quantified and principal component analysis performed on these variable genes. T-stochastic nearest neighbour embedding (t-SNE) dimensional reduction was performed on the top 15 principal components to obtain embeddings for individual cells. GSVA was performed using the 15 genes used for HIF score calculation as above. An oxidative stress score for each cell was calculated using GSVA based on a list of genes upregulated in IPF cell populations (*ABCC1, CDKN2D, FES, GCLC, GCLM, GLRX2, HHEX, IPCEF1, JUNB, LAMTOR5, LSP1, MBP, MGST1, MPO, NDUFA6, PFKP, PRDX1, PRDX2, PRDX4, PRNP, SBNO2, SCAF4, SOD1, SOD2, RXN1, TXN, TXNRD1*) from Gene Set: HALLMARK_REACTIVE_OXYGEN_SPECIES_PATHWAY (M5938). All parameters and variables can be found in the accompanying file (*Source code 3*). Upregulated oxidative stress genes were those whose expression was higher in IPF populations than control. Calculated GSVA scores

were mapped onto t-SNE plots. Student's t-test was used to calculate statistical differences between GSVA scores of the different cellular populations.

## 3D in vitro model of fibrosis

Culture was performed as previously described (*Jones et al., 2018*). Briefly, peripheral lung fibroblasts were obtained as outgrowths from surgical lung biopsy tissue of patients (n = 3 donors) who were subsequently confirmed with a diagnosis of IPF. All primary cultures were tested and free of myco-plasma contamination. The fibroblasts were seeded in Transwell inserts in DMEM containing 10% FBS. After 24 hr, the media was replaced with DMEM/F12 containing 5% FBS, 10 µg/ml L-ascorbic acid-2-phosphate, 10 ng/ml EGF, and 0.5 µg/ml hydrocortisone with or without 50 µM or 250 µM IOX2 and/or 1 mM DM-NOFD (*McDonough et al., 2005*), as indicated; each experiment included a vehicle control (0.2% DMSO). TGF-β1 (3 ng/mL) was added to the cultures, and the medium replenished three times per week. After 2 weeks spheroids were lysed for western blotting. After 6 weeks, the spheroids were either snap frozen for parallel-plate compression testing, analysis of cross-linking, and histochemical staining, or fixed using 4% paraformaldehyde for histochemistry or 3% glutaraldehyde in 0.1 M cacodylate buffer at pH 7.4 for electron microscopy.

## Reverse transcription quantitative polymerase chain reaction (RTqPCR)

RTqPCR was performed as previously described (*Yao et al., 2019*; *Conforti et al., 2020*; *Hill et al., 2019b*). Primers and TaqMan probe sets were obtained from Primer Design, Southampton, UK (*LOXL2, COL1A1, Col3A1, PLOD2*), ThermoFisher Scientific, Reading, UK (*HIF1A* [HIF1α], *EPAS1* [HIF2α], *ARNT* [HIF1β]), and Qiagen, Manchester, UK (QuantiTect Primer Assays, *HIF1A, EPAS1, ARNT, LOXL2, PLOD2, CA9, ACTB*).

## Western blotting

Fibroblasts were lysed using 2 x Laemmli SDS sample buffer or urea buffer (8 M Urea, 1 M Thiourea, 0.5% CHAPS, 50 mM DTT, and 24 mM Spermine). Western blotting of cellular lysates was performed for β-actin (1:100.000, Sigma-Aldrich, Poole, UK), LOXL2 (1:1000, R&D Systems, Abingdon, UK), HIF1α (1:1000, BD Biosciences, Wokingham, UK), FIH (1:200, mouse monoclonal 162 C) (*Wang et al., 2018*), β-tubulin (1:1000, Cell Signaling Technology, London, UK), HIF1 β (1:1000, Cell Signaling Technology), p-Smad2/3 (1:1000, Cell Signaling Technology), p-ERK (1:1000, Cell Signaling Technology), active β-catenin (1:1000, Cell Signaling Technology). Immunodetected proteins were identified using the enhanced chemiluminescence system (Clarity Western Blotting ECL Substrate, Bio-Rad Laboratories Ltd, Watford, UK) or Odyssey imaging system (LI-COR), and evaluated by ImageJ 1.42q software (National Institutes of Health).

## Immunofluorescence staining

Cells were fixed with 4% paraformaldehyde followed by permeabilisation and staining with primary antibodies for LOXL2 (1:100, R&D Systems), PLOD2 (1:100, Proteintech) and tetramethylrhodamine (TRITC)-conjugated Phalloidin (1:1000, Millipore UK Limited, Watford, UK). The secondary antibodies used were Alexafluor 488 and 647 (1:1000, BioLegend UK Ltd, London, UK). Cell nuclei were counterstained with 4',6-Diamidino-2-Phenylindole, Dihydrochloride (DAPI) (1:1000, Millipore UK Limited, Watford, UK). Cells were imaged using an inverted confocal microscope (Leica TCS-SP5 Confocal Microscope, Leica Microsystems).

## Immunohistochemistry

Control or IPF lung tissues (n = 3 donors) were fixed and embedded in paraffin wax; tissue sections (4 µm) were processed and stained as previously described (*Yao et al., 2019*; *Hill et al., 2019b*). Briefly, the tissue sections were de-waxed, rehydrated and incubated with 3% hydrogen peroxide in methanol for 10 min to block endogenous peroxidase activity. Sections were then blocked with normal goat serum and incubated at room temperature with a primary antibody against CA-IX (1:500, Novus Biologicals, Cambridge, UK) or HIF1α (1:500, Cayman Chemical, Michigan, USA), followed by a biotinylated secondary antibody (1:500, Vector Laboratories Ltd., UK); antibody binding was detected using streptavidin-conjugated horse-radish peroxidase and visualised using DAB before counterstaining with Gill's Haematoxylin. Images were acquired using an Olympus Dotslide Scanner VS110.

## Picrosirius red collagen area quantitation

Sample sections, stained with Picrosirius Red as previously described (*Jones et al., 2018*), were imaged under polarised light and 10 areas were selected at random for each condition (5 each from two donors). Images of dimension 1498 × 1221 pixels with a pixel size of 0.14 μm x 0.14 μm were taken using Olympus Olyvia software and converted through ImageJ to binary RGB images using pre-determined threshold levels (low 25, high 255) to demonstrate areas of collagen fibres only, as previously described (*Hadi et al., 2011*). The proportion of area composed of collagen fibres within total sample area was then calculated.

Protein, hydroxyproline and collagen cross-link assays performed as previously described (*Jones et al., 2018*).

Parallel plate compression testing: performed as previously described (*Jones et al., 2018*).

Transmission electron microscopy: performed as previously described (*Jones et al., 2018*).

Atomic force microscopy nanoindentation imaging of individual non-hydrated collagen fibrils: performed as previously described (*Jones et al., 2018*).

## Statistics

Statistical analyses were performed in GraphPad Prism v7.02 (GraphPad Software Inc, San Diego, CA) unless otherwise indicated. No data were excluded from the studies and for all experiments, all attempts at replication were successful. For each experiment, sample size reflects the number of independent biological replicates and is provided in the figure legend. Normality of distribution was assessed using the D'Agostino-Pearson normality test. Statistical analyses of single comparisons of two groups utilised Student's t-test or Mann-Whitney U-test for parametric and non-parametric data respectively. Where appropriate, individual t-test results were corrected for multiple comparisons using the Holm-Sidak method. For multiple comparisons, one-way analysis of variance (ANOVA) with Dunnett's multiple comparison test or Kruskal-Wallis analysis with Dunn's multiple comparison test were used for parametric and non-parametric data, respectively. Results were considered significant if $p < 0.05$, where $*p < 0.05$, $**p < 0.01$, $***p < 0.001$, $****p < 0.0001$.

## Acknowledgements

This project was supported by Medical Research Council (MR/S025480/1), the Wellcome Trust (100638/Z/12/Z), an Academy of Medical Sciences/the Wellcome Trust Springboard Award [SBF002\1,038], and the AAIR Charity. CJB and LSND acknowledge the support of the NIHR Southampton Biomedical Research Centre. LY was supported by China Scholarship Council. YZ was supported by an Institute for Life Sciences PhD Studentship. FC was supported by Medical Research Foundation [MRF-091–0003-RG-CONFO]. ML was supported by a BBSRC Future Leader Fellowship [BB/PO11365/1] and a NIHR Southampton Biomedical Research Centre Senior Research Fellowship. We thank Carine Fixmer, Maria Lane, Benjamin Johnson, and the nurses of the Southampton Biomedical Research Unit for their help in the collection of human samples, supported by the Wessex Clinical Research Network and the National Institute of Health Research, UK. We also thank Dr. Tammie Bishop (University of Oxford) for her technical support in IHC and Prof Sir Peter Ratcliffe (University of Oxford) for the FIH antibody (mouse monoclonal 162 C) and the UAS-luc/GAL4DBD-HIF1αCAD binary reporter system.

## Additional information

### Funding

| Funder | Grant reference number | Author |
| --- | --- | --- |
| Wellcome Trust | 100638/Z/12/Z | Mark Jones |
| Medical Research Council | MR/S025480/1 | Yihua Wang |
| Academy of Medical Sciences | SBF002\1038 | Yihua Wang |

| Funder | Grant reference number | Author |
|---|---|---|

The funders had no role in study design, data collection and interpretation, or the decision to submit the work for publication.

## Author contributions

Christopher J Brereton, Conceptualization, Formal analysis, Funding acquisition, Investigation, Methodology, Writing – original draft, Writing – review and editing; Liudi Yao, Yilu Zhou, Formal analysis, Investigation, Methodology, Writing – original draft, Writing – review and editing; Elizabeth R Davies, Formal analysis, Investigation, Methodology, Project administration, Writing – review and editing; Milica Vukmirovic, Formal analysis, Investigation, Methodology, Resources, Writing – original draft, Writing – review and editing; Joseph A Bell, Formal analysis, Methodology, Resources, Writing – original draft, Writing – review and editing; Siyuan Wang, Investigation, Writing – review and editing; Robert A Ridley, Formal analysis, Investigation, Methodology, Project administration, Writing – original draft, Writing – review and editing; Lareb SN Dean, Formal analysis, Investigation, Methodology, Writing – review and editing; Orestis G Andriotis, Franco Conforti, Investigation, Methodology, Writing – review and editing; Lennart Brewitz, Rob M Ewing, Naftali Kaminski, Luca Richeldi, Atul Bhaskar, Christopher J Schofield, Resources, Writing – review and editing; Soran Mohammed, Aurelie Fabre, Matthew Loxham, Investigation, Methodology, Resources, Writing – review and editing; Timothy Wallis, Ali Tavassoli, Investigation, Resources, Writing – review and editing; Aiman Alzetani, Benjamin G Marshall, Sophie V Fletcher, Philipp J Thurner, Methodology, Resources, Writing – review and editing; Donna E Davies, Conceptualization, Formal analysis, Funding acquisition, Investigation, Methodology, Supervision, Writing – original draft, Writing – review and editing; Yihua Wang, Mark G Jones, Conceptualization, Formal analysis, Funding acquisition, Investigation, Methodology, Project administration, Resources, Supervision, Writing – original draft, Writing – review and editing

## Author ORCIDs

Christopher J Brereton http://orcid.org/0000-0001-8302-702X
Elizabeth R Davies http://orcid.org/0000-0002-8629-8324
Lareb SN Dean http://orcid.org/0000-0002-8703-9236
Soran Mohammed http://orcid.org/0000-0002-3882-6129
Benjamin G Marshall http://orcid.org/0000-0003-0946-0399
Naftali Kaminski http://orcid.org/0000-0001-5917-4601
Christopher J Schofield http://orcid.org/0000-0002-0290-6565
Matthew Loxham http://orcid.org/0000-0001-6459-538X
Donna E Davies http://orcid.org/0000-0002-5117-2991
Yihua Wang http://orcid.org/0000-0001-5561-0648
Mark G Jones http://orcid.org/0000-0001-6308-6014

## Ethics

Human subjects: All human lung experiments were approved by the Southampton and South West Hampshire and the Mid and South Buckinghamshire Local Research Ethics Committees (ref 07/H0607/73), and all subjects gave written informed consent.

## Decision letter and Author response

Decision letter https://doi.org/10.7554/eLife.69348.sa1
Author response https://doi.org/10.7554/eLife.69348.sa2

# Additional files

## Supplementary files

• Supplementary file 1. Semiquantitative analysis of mRNA expression & Fibroblast demographic details. (a) Semiquantitative analysis of *LOX2* and *PLOD2* mRNA expression identified by RNAscope in situ hybridization in cell subtypes in IPF lung tissue (n = 7 donors). FF, fibroblast focus. (b) Fibroblast donor demographic details.

• Supplementary file 2. Short interfering RNA (siRNA) oligo sequences.

• Transparent reporting form

• Source code 1. Source code for RNAseq analyses for *Figure 6a, b*.

- Source code 2. Source code for RNAseq analyses for *Figure 7a, b*.
- Source code 3. Source code for RNAseq analyses for *Figure 8* and *Figure 8—figure supplement 1*.

### Data availability

All data generated or analysed during this study are included in the manuscript and supporting files.

The following previously published datasets were used:

| Author(s) | Year | Dataset title | Dataset URL | Database and Identifier |
|---|---|---|---|---|
| Habermann AC, Gutierrez AJ, Bui LT, Winters NI, Calvi CL, Peter L, Chung M, Taylor CJ, Yahn SL, Jetter C, Raju L, Roberson J, Ding G, Wood L, Sucre JM, Richmond BW, Serezani AP, McDonnell WJ, Mallal SB, Bacchetta MJ, Shaver CM, Ware LB, Bremner R, Walia R, Blackwell TS, Banovich NE, Kropski JA | 2019 | Single-cell RNA-sequencing reveals profibrotic roles of distinct epithelial and mesenchymal lineages in pulmonary fibrosis | https://www.ncbi.nlm.nih.gov/geo/query/acc.cgi?acc=GSE135893 | NCBI Gene Expression Omnibus, GSE135893 |
| Lindahl GE, Stock CJ, Shi-Wen X, Nicholson AG, Dusmet ME, Bou-Gharios G, Abraham DJ, Denton CP, Wells AU, Renzoni EA | 2013 | Expression data from fibroblasts cultured from normal and fibrotic human lung tissue | https://www.ncbi.nlm.nih.gov/geo/query/acc.cgi?acc=GSE40839 | NCBI Gene Expression Omnibus, GSE40839 |
| Thannickal VJ, Chanda D | 2017 | Developmental programming in Idiopathic pulmonary fibrosis (IPF) | https://www.ncbi.nlm.nih.gov/geo/query/acc.cgi | NCBI Gene Expression Omnibus, GSE73854 |
| Vukmirovic M, Jones MG, Kaminski N | 2021 | Spatial transcriptome profiling identifies CREB1 as a regulator of core transcriptional programs in idiopathic pulmonary fibrosis | https://www.ncbi.nlm.nih.gov/geo/query/acc.cgi?acc=GSE169500 | NCBI Gene Expression Omnibus, GSE169500 |

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

# Appendix 1

### Appendix 1—key resources table

| Reagent type (species) or resource | Designation | Source or reference | Identifiers | Additional information |
|---|---|---|---|---|
| transfected construct (human) | GAL4DBD-HIF1αCAD (residues 652–826) for HIF1α CAD reporter assay | Ratcliffe lab (University of Oxford) *Coleman et al., 2007* | | |
| transfected construct (Homo-sapiens) | UAS-luc reporter for HIF1α CAD reporter assay | Ratcliffe lab (University of Oxford) *Coleman et al., 2007* | | |
| transfected construct (Homo-sapiens) | Plasmid for Dual-Luciferase Reporter Assay | Promega | phRL-CMV | |
| transfected construct (Homo-sapiens) | siRNA to human HIF1AN (FIH) | Dharmacon/ Thermo Fisher Scientific | MU-004073-02-0002 | |
| transfected construct (Homo-sapiens) | siRNA to human HIF1A (HIF1α) | Dharmacon/ Thermo Fisher Scientific | MU-00401805-05-0002 | |
| transfected construct (Homo-sapiens) | siRNA to human EPAS1(HIF2α) | Dharmacon/ Thermo Fisher Scientific | MU-004814-01-0002 | |
| transfected construct (Homo-sapiens) | siRNA to human ARNT (HIF1β) | Dharmacon/ Thermo Fisher Scientific | MU-007207-01-0002 | |
| transfected construct (Homo-sapiens) | siGENOME RISC-Free | Dharmacon/ Thermo Fisher Scientific | D-001220-01-05 | |
| Antibody | Anti-CAIX (Rabbit polyclonal) | Novus Biologicals | Cat. #: NB100-417 | IHC 1:500 |
| Antibody | Anti-HIF1A (Rabbit polyclonal) | Cayman Chemical | Cat. #: 10006421 | IHC 1:500 |
| antibody | anti-human HIF1α (Mouse polyclonal IgG1$_k$) | BD Biosciences | Cat #:610,958 | WB (1:1000) |
| antibody | Anti-HIF1β (Rabbit polyclonal) | Cell Signaling Technology | Cat #:5,537 | WB (1:1000) |
| antibody | Anti-phospho-Smad2 (Rabbit polyclonal) | Cell Signaling Technology | Cat #: 3,104 | WB (1:1000) |
| antibody | anti-β-tubulin (Mouse polyclonal) | Cell Signaling Technology | Cat #: 86,298 | WB (1:1000) |
| antibody | anti-PLOD2 (Mouse monoclonal IgG$_{2B}$) | R&D Systems | Cat #: MAB4445 | WB (1:500) |
| antibody | Anti- human LOXL2 (Goat polyclonal) | R&D Systems | Cat #: AF2639 | WB (1:1000) IF (1:100) |
| antibody | anti-human FIH (Mouse monoclonal 162 C) | Ratcliffe lab (University of Oxford) *Stolze et al., 2004* | | WB (1:200) |
| Antibody | Anti-P-ERK (polyclonal rabbit Thr202/ Tyr204) | Cell Signalling Technology | Cat #: 9,101 | WB (1:1000) |
| Antibody | Anti-P-SMAD2/3 (Rabbit polyclonal Ser465/467) | Cell Signalling Technology | Cat #: 8,828 | WB (1:1000) |

*Appendix 1 Continued on next page*

*Appendix 1 Continued*

| Reagent type (species) or resource | Designation | Source or reference | Identifiers | Additional information |
|---|---|---|---|---|
| Antibody | IRDye 800CW Donkey anti-Goat IgG Secondary Antibody | LI-COR Biosciences | Cat #: 926–32214 | WB (1:5000) |
| Antibody | IRDye 800CW Goat anti-Rabbit IgG Secondary Antibody | LI-COR Biosciences | Cat #: 926–32211 | WB (1:5000) |
| Antibody | IRDye 680LT Goat anti-Mouse IgG Secondary Antibody | LI-COR Biosciences | Cat #: 926–68020 | WB (1:5000) |
| Antibody | Anti-non-phospho (active) β-catenin (Rabbit monoclonal IgG) | Cell Signalling Technology | Cat. #: 8,814 S | WB (1:1000) |
| Antibody | Anti-mouse IgG HRP-linked whole antibody | Life Sciences | Cat. #: NXA931 | WB (1:1000) |
| Antibody | Anti-goat Immunoglobulins/HRP (affinity isolated) | Dako | Cat. #: P0449 | WB (1:1000) |
| Antibody | Anti-goat IgG H&L (Alexa Fluor 647) | Abcam | Cat. #: Ab150131 | ICC 1:100 |
| sequence-based reagent | Human *HIF1A* (HIF1α) | Qiagen | QuantiTect PCR primers Cat #: QT00083664 | |
| sequence-based reagent | Human *EPAS1* (HIF2α) | Qiagen | QuantiTect PCR primers Cat #: QT00069587 | |
| sequence-based reagent | Human *ARNT* (HIF1β) | Qiagen | QuantiTect PCR primers Cat #: QT00023177 | |
| sequence-based reagent | Human *ACTB*(β-actin) | Qiagen | QuantiTect PCR primers Cat #: QT01680476 | |
| Sequence-based reagent | LOXL2 | Primer Design | | |
| Sequence-based reagent | PLOD2 | Primer Design | | |
| Sequence-based reagent | COL1A1 | Primer Design | | |
| Sequence-based reagent | COL3A1 | Primer Design | | |
| Peptide, recombinant protein | Recombinant Human TGF-beta 1 Protein | R&D Systems | Cat. #: 240-B-010 | |
| Peptide, recombinant protein | Recombinant Human EGF GMP Protein | R&D Systems | Cat. #: 236-GMP-200 | |
| Commercial assay or kit | RNAscope 2.5 HD Duplex Assay | Advanced Cell Diagnostics | Cat. #: 322,430 | |
| Commercial assay or kit | RNAscope probe-Hs-LOXL2-C1 | Advanced Cell Diagnostics | Cat. #: 311,341 | |
| Commercial assay or kit | RNAscope probe-Hs-PLOD2-C2 | Advanced Cell Diagnostics | Cat. #: 547761-C2 | |

*Appendix 1 Continued on next page*

*Appendix 1 Continued*

| Reagent type (species) or resource | Designation | Source or reference | Identifiers | Additional information |
|---|---|---|---|---|
| Commercial assay or kit | MicroVue Bone PYD EIA | Quidel | Cat. #: 8,010 | |
| Commercial assay or kit | Hydroxyproline Assay Kit | Merck | Cat. #: MAK008 | |
| Commercial assay or kit | Total Protein Assay | QuickZyme Biosciences | Cat. #: QZBtotprot | |
| Commercial assay or kit | Picro Sirius Red Stain Kit (Connective Tissue Stain) | Abcam | Cat. #: Ab150681 | |
| commercial assay or kit | Lipofectamine 3,000 | Thermo Fisher Scientific | Cat. #: L3000008 | |
| commercial assay or kit | Lipofectamine RNAiMAX | Thermo Fisher Scientific | Cat. #: 13778–075 | |
| commercial assay or kit | Dual-Luciferase Reporter Assay System | Promega | Cat. #: E1910 | |
| commercial assay or kit | QuantiNova SYBR Green RT-PCR kits | Qiagen | Cat. #: 208052 | |
| chemical compound, drug | Dimethyloxaloylglycine (DMOG) | Sigma Aldrich | Cat #: D3695 CAS: 89464-63-1 | |
| chemical compound, drug | *N*-[[1,2-Dihydro-4-hydroxy-2-oxo-1-(phenylmethyl)–3-quinolinyl]carbonyl]-glycine (IOX2) | Selleck Chemicals | Cat #: S2919 CAS: 931398-72-0 | |
| chemical compound, drug | Dimethyl *N*-oxalyl-D-phenylalanine (DM-NOFD) | Schofield lab (University of Oxford) **McDonough et al., 2005** | | |
| chemical compound, drug | DMSO | Sigma Aldrich | Cat #: 276,855 CAS: 67-68-5 | |
| chemical compound, drug | T-hydro (tert-butyl hydroperoxide) | Sigma Aldrich | Cat #: 19,999 CAS: 75-91-2 | 20 µM (fresh prepared) |

