## [Editor Report]

The reviewers found your manuscript of broad interest to researchers interested in lung biology, as the study builds upon the previous original work of the authors, by identifying a pathway that regulates collagen nanostructure and stiffness in lung fibrosis and demonstrating that this pathway it is independent of pathways regulating collagen synthesis. They also valued the elegant analysis you performed to validate the specificity of experimental finding, and demonstrate that HIF activation is required for the increased tissue stiffness associated with fibrosis.

---

## [Decision Letter]

**Decision letter after peer review:**

Thank you for submitting your article "Pseudohypoxic HIF pathway activation dysregulates collagen structure-function in human lung fibrosis" for consideration by *eLife*. Your article has been reviewed by 3 peer reviewers, and the evaluation has been overseen by a Reviewing Editor and Paul Noble as the Senior Editor. The following individual involved in review of your submission has agreed to reveal their identity: Gianni Carraro (Reviewer #3).

The reviewers found your manuscript of broad interest to researchers interested in lung biology and especially to those focused on the pathogenesis and therapy of lung fibrosis, and find the data strong and convincing. They recognize that this study builds upon the previous original work of the authors, by identifying a pathway that regulates collagen nanostructure and stiffness in lung fibrosis and demonstrating that this pathway it is independent of pathways regulating collagen synthesis. They also valued the elegant analysis you performed to validate the specificity of experimental finding, and the demonstration that HIF activation is required for the increased tissue stiffness associated with fibrosis. They also raise several important points and constructive suggestions we would like you to address before we can make decision about the suitability of your manuscript for publication in *eLife*.

Essential revisions:

1) Although the authors consider the fact that LOX and LOXL1 are also increased, and justify their focus on LOXL2 by its correlation with PLOD2 expression, it may not be justified to completely ignore LOX and LOXL1. The reported data do not rule out a role for LOX and LOXL1.

2) The authors need to show direct involvement of LOXL2 and PLOD2, e.g. using siRNA to revert the stiffness phenotype. They also should show co-localization of LOXL2 and PLOD2 in areas of COL1A1 and COL 3A deposition), as there is a synergistic effect between TGF β and HIF pathways in IPF foci.

3) Measurements of the fiber width, length and density quantification would increase the confidence into the morphological differences shown in picrosirius red staining images.

4) The authors should show criteria and characterization of the healthy and diseased donors of primary lung fibroblasts used in the experiments.

5) The mRNA data should be confirmed with immunostaining/WB as detailed in the reviews.

*Reviewer #2 (Recommendations for the authors):*

Figure 1h. PLOD 2 should be counterstained with LOXL2 in tissues with IPF foci and show high expression level versus non foci area in same samples and healthy lung tissue samples. Lower magnification images (together with high magnification quadrants) should be presented to show that overexpression is more than in few isolated spots.

Figure 2. b, c, e show some discrepancies in levels of LOXL2. Panel b has the highest level of LOXL2 mRNA at 72hrs, while the western blot in c showed only minimal increase of LOXL2 protein at same time point (72 hrs) compared to other conditions. What happens at 24 and 48 hrs at protein level? In panel E, the immunostaining for LOXL2 at 24hrs shows a notable difference with controls. This is not what would be expected based on the WB data at 72hrs. Authors needs to show WB and immunostaining data at same time point to support the quality of their data using different methodologies and support their conclusions. PLOD 2 immunostaining needs to be included in this figure to support the role of both PLOD2 and LOXL2 in the fibrotic phenotype.

Figure 3. If HIF levels are supposedly higher in IPF fibroblasts, authors should compare them to the levels of HIF in healthy fibroblasts and show how much they can be reduced in both samples (healthy versus IPF) by siRNA.

Figure 4A- B. These experiments show a significant synergistic effect of TGFbeta with DMOG and IOX2 on the protein levels of PLOD2, and none on LOXL2 levels. The data don't support the same magnitude effects shown in Figure 2 C. The authors need to do the same experiment as Figure 4 A and 4B using healthy and IPF fibroblasts (3 healthyversus 3 IPF donors) and compare them at 72 hrs as in Figure 2C, and clarify why 48hrs time point was used here. TGFbeta + DMOG appear to decrease the protein levels of LOXL2 compared to DMOG alone. These data are in disagreement with mRNA levels in Figure 4C. The authors need to explain in the text this difference and why.

Figure 5. To show a direct effect of PLOD2 and LOXL2 on pyridoline cross-linking and ECM stiffness, authors can do the same experiment as Figure 5 A and B with siRNA for PLOD2 and LOXL2 after IOX2 treatment and see whether this can revert the phenotype.

Figure 7. The claim of HIF/LOXL2/PLOD 2 colocalization needs to be supported by more images (quality and quantity). PLOD/LOXL2 constating is the same of Figure 2. Authors should provide co-staining of PLOD2/LOXL2 and HIF/CAIX in the same slide. If hard to obtain, PLOD2/LOXL2 with HIF and PLOD2/LOXL2 with CAIX costaining and show lower and higher magnification images and no few single events as in the actual figure.

*Reviewer #3 (Recommendations for the authors):*

Overall this manuscript will be an important contribution to the field of lung fibrosis. The experiments are well executed and include adequate number of biological replicates for statistical analysis. One aspect I would consider modifying is the interpretation that the identified effect of HIF is acting in non-hypoxic conditions. At least from the data presented it doesn't look obvious that this is the case. If there are no compelling evidence for this claim I would modify the narrative and include this as a possibility in the discussion.

Below are some points for improvement of the manuscript:

Figure 1: The quantification of the correlation of expression of PLOD2 and LOXL2 is properly executed and include comparison with other cross-linking enzyme to show specificity of the result. Figure 1h corroborate this result by showing co-localization by ISH, and it would be helpful to add an high magnification of a negative region to show specificity of the staining.

Figure 7: Try to show co-localization from serial sections. This does not allows to determine if the same or adjacent cells were probed. A more definitive way to achieve this would be to perform ISH with multiple fluorescent probes, using the same technique (RNAscope) that here was used with a chromogenic approach that limit the co-localization to two transcripts. In alternative RNAscope + immunofluorescence could also be combined for the same purpose.

The single cell data displayed in Figure 8 could also be used to corroborate cellular co-expression of PLOD2/LOXL2 and HIF1a in addition to the signature score that are provided.

Figure 8: From the code description it seems that a dimensional reduction was created first, followed by dataset subsetting to include only 'Control' and 'IPF' in the final figure. Did the authors tried to create a dimensional reduction after subsetting the 'Kropski' dataset and see if this can improve the segregation of the clusters of interest?

The color used in the tSNE plots in Figure 8a, 8c, S6a may be difficult to distinguish for color blind people. I suggest the use of alternative color palettes. For R, several packages are available for this purpose. The 'viridis' package for example use color blind accessible color scales.

The 'Cell Type' naming on top of the tSNE map is difficult to read, I suggest to adjust the position or to have it as a side legend.

---

## [Author Response]

Essential revisions:1) Although the authors consider the fact that LOX and LOXL1 are also increased, and justify their focus on LOXL2 by its correlation with PLOD2 expression, it may not be justified to completely ignore LOX and LOXL1. The reported data do not rule out a role for LOX and LOXL1.

We thank the reviewer for highlighting this comment which we have now further considered within the discussion (Page 14 Line 302 – Page 15 Line 315). The LOX(L) enzymes play key roles in the process of fibrillar collagen production and are tightly regulated in normal development and under physiological conditions (Trackman PC. Expert Opinion on Therapeutic Targets 2016;20:935–945). In our LCMD RNA-Seq analyses LOXL2 was the most highly expressed LOX/LOXL family member as well as the only LOX/LOXL member which correlated with PLOD2 expression, whilst in our previous work investigating collagen structure-function dysregulation in human lung fibrosis (eLife 2018;7:e36354) we identified that gene expression of LOXL2 was significantly increased in IPF tissue when compared to age-matched control lung tissue. Furthermore, using a small molecule LOXL inhibitor in our 3D model of fibrosis, we previously identified a significant reduction in mature pyridinoline cross-links using a concentration which completely inhibits LOXL2 but has minimal effects on LOX and LOXL1 (eLife 2018;7:e36354). This is consistent with previous reports that LOXL2 has key pathologic roles in cancer and fibrosis (Barker et al., Nature Reviews Cancer 2012;12:540–552; Barry-Hamilton et al., Nature Medicine 2010;16:1009–1017). Therefore, whilst we acknowledge within the discussion that our studies do not exclude a potential role for LOX and LOXL1 in human lung fibrosis, here we focussed upon the LOXL2/PLOD2 relationship.

2) The authors need to show direct involvement of LOXL2 and PLOD2, e.g. using siRNA to revert the stiffness phenotype. They also should show co-localization of LOXL2 and PLOD2 in areas of COL1A1 and COL 3A deposition), as there is a synergistic effect between TGF β and HIF pathways in IPF foci.

As requested, we tested small interfering RNA (siRNA) knockdown against *LOXL2* within our long term 6 week 3D model of fibrosis. Unfortunately, *LOXL2* knockdown was not sustained throughout the 6 week culture (Author response image 1), although the knockdown was sufficient to significantly reduce tissue stiffness in the presence of IOX2 (Author response image 1). This is a finding consistent with our previous study using a small molecule LOXL inhibitor in our 3D model of fibrosis (eLife 2018;7:e36354) where we demonstrated a significant reduction in tissue stiffness using a concentration which completely inhibits LOXL2/3 but not other LOX(L) family members. We did not feel it was appropriate to include these data as we were unable to assess the impact of long-term silencing upon tissue stiffness, therefore we acknowledge within the discussion that an approach such as CRISPR would be required for persistent gene silencing and that this would be a potential area for future study (Page 15 Lines 313-315). If the Reviewers felt that including these data was helpful we would be willing to include them within the Supplementary Results.

**Author response image 1. sa2fig1:** 

In our previous studies, through human lung tissue analyses and in vitro studies we identified that lung fibrosis tissue stiffness is significantly determined by pyridinoline collagen cross-linking. PLOD2 encodes the lysyl hydroxylase 2 (LH2) enzyme, which is responsible for the hydroxylation of lysine residues in fibrillar collagen telopeptides. Telopeptide hydroxylysines are essential for the hydroxyallysine pathway of cross-linking, and this is required for the production of all mature lysylpyridinoline and hydroxylysylpyridinoline cross-links in extracellular collagen fibrils (van der Slot AJ et al., J Biol Chem. 2003;278:40967–72 and Gistelinck et al., J Bone Miner Res. 2016; 31: 1930–1942), with promotion of PLOD2 expression increasing tissue stiffness and pyridinoline cross-links (Chen et al., J Clin Invest. 2015;125:1147-62). Thus PLOD2 actitivty is essential for the presence of pyridinoline (hydroxylysine aldehyde-based) cross-links which we and others have demonstrated significantly determine tissue stiffness. In view of these considerations and the problems of maintaining silencing of LOXL2, we did not investigate siRNA PLOD2 knockdown within our current studies.To demonstrate co-localization of LOXL2 and PLOD2 in areas of fibrillar collagen (COL1A1 and COL3A1) deposition we performed Masson’s trichrome staining for collagen fibres on serial sections adjacent to those used for LOXL2/PLOD2 RNAScope, clearly identifying that LOXL2 and PLOD2 co-localize within areas of fibrillar collagen deposition (new Figure 1—figure supplement 1h and Page 6, Lines 94-95).

3) Measurements of the fiber width, length and density quantification would increase the confidence into the morphological differences shown in picrosirius red staining images.

As requested we have performed density (surface area quantification) of the Picrosirius red staining, identifying no difference in Picrosirius red surface area between conditions (new Figure 5—figure supplement 1c). In our analyses we performed Picrosirius red for qualitative comparison between conditions, however we did not perform detailed quantitative analyses for metrics such as collagen fiber width and length as, in our experience and those of others (Nazac et al., Microsc Res Tech. 2015;78:723-30. Latouf et al., Histochem Cytochem. 2014 Oct;62:751-8), there are potential limitations to the application of Picrosirius red imaging for quantitative fibrillar collagen analyses depending upon the tissue studied and imaging approach. Whilst picrosirius red staining is a well established histological detection method for binding to cationic collagen fibers, the utility for detailed quantitative assessment of collagen structure is less established and a requirement for validation for any specific tissue is recognised (Drifka et al., J Histochem Cytochem. 2016; 64: 519–529). We therefore utilised an accepted gold standard of transmission electron microscopy for our quantitative fibrillar collagen analyses.

4) The authors should show criteria and characterization of the healthy and diseased donors of primary lung fibroblasts used in the experiments.

As requested we have now included these data (new Supplementary File 1b). Across all experiments we used a total of 7 IPF lung fibroblast lines and 5 healthy lung fibroblast lines which were age, sex, and smoking history matched.

5) The mRNA data should be confirmed with immunostaining/WB as detailed in the reviews.

As requested we have included additional protein expression data. Within our 72 hour time course experiment (Figure 2a-d) of profibrogenic stimuli the 24 and 48hr protein expression levels of PLOD2 and LOXL2 were requested to complement the included mRNA data and 72hr protein expression (Figure 2a,b,c). We have now included western blotting data for LOXL2 and PLOD2 at each point within the time course (24,48 and 72hrs) (new Figure 2—figure supplement 1b) in the presence or absence of TGFβ1 or the hypoxia mimetic DMOG. Together our protein and gene expression data consistently identify that DMOG most strongly upregulates both PLOD2 and LOXL2 expression at each time point studied. To complement the western blot analyses of protein expression we performed additional immunofluorescent staining for PLOD2 and LOXL2 in the presence or absence of TGFb or the hypoxia mimetics IOX2, and DMOG (*new Figure 2e*), confirming by immunofluorescent staining an increase in intracellular LOXL2 and PLOD2 expression following DMOG or IOX2 treatment in comparison to treatment with TGFβ1.

Reviewer #2 (Recommendations for the authors):Figure 1h. PLOD 2 should be counterstained with LOXL2 in tissues with IPF foci and show high expression level versus non foci area in same samples and healthy lung tissue samples. Lower magnification images (together with high magnification quadrants) should be presented to show that overexpression is more than in few isolated spots.

Co-localisation is present across fibroblast foci of diagnostic lung tissue for IPF patients (n=7) studied. The pattern of in situ hybridization was assessed by an expert histopathologist (AF) who confirmed consistent co-expression, and we have now additionally included a supplementary table of semi-quantitative analysis of PLOD2 and LOXL2 RNAScope expression within cell types in IPF tissue confirming expression is greatest with fibroblast foci (Supplementary File 1a) Additional examples including lower magnification images as well as high magnification images are also provided (new Figure 1-figure supplement 1f-h).

Figure 2. b, c, e show some discrepancies in levels of LOXL2. Panel b has the highest level of LOXL2 mRNA at 72hrs, while the western blot in c showed only minimal increase of LOXL2 protein at same time point (72 hrs) compared to other conditions. What happens at 24 and 48 hrs at protein level? In panel E, the immunostaining for LOXL2 at 24hrs shows a notable difference with controls. This is not what would be expected based on the WB data at 72hrs. Authors needs to show WB and immunostaining data at same time point to support the quality of their data using different methodologies and support their conclusions. PLOD 2 immunostaining needs to be included in this figure to support the role of both PLOD2 and LOXL2 in the fibrotic phenotype.

Please see the response to Essential Revision 5.

Figure 3. If HIF levels are supposedly higher in IPF fibroblasts, authors should compare them to the levels of HIF in healthy fibroblasts and show how much they can be reduced in both samples (healthy versus IPF) by siRNA.

Elevated levels of HIF1 and HIF2 in IPF fibroblasts under normoxic conditions have recently been reported (Aquino-Galvez, A. et al., Respir Res 20, 130 (2019)), and so in our studies we investigated possible mechanism for this. We focussed upon assessment of FIH activity, identifying using a luciferase reporter system that activity of FIH is decreased in lung fibroblasts from patients with IPF, so identifying a potential mechanism for increased HIF pathway activation in IPF. Consistent with this we identified in publicly available datasets that cultured fibroblasts from patients with a usual interstitial pneumonia pattern of fibrosis have a significantly increased HIF score compared to cultured control fibroblasts.

Figure 4A- B. These experiments show a significant synergistic effect of TGFbeta with DMOG and IOX2 on the protein levels of PLOD2, and none on LOXL2 levels. The data don't support the same magnitude effects shown in Figure 2 C. The authors need to do the same experiment as Figure 4 A and 4B using healthy and IPF fibroblasts (3 healthyversus 3 IPF donors) and compare them at 72 hrs as in Figure 2C, and clarify why 48hrs time point was used here. TGFbeta + DMOG appear to decrease the protein levels of LOXL2 compared to DMOG alone. These data are in disagreement with mRNA levels in Figure 4C. The authors need to explain in the text this difference and why.

As TGFβ1 strongly induced major collagen fibrillogenesis genes whilst HIF pathways most strongly induced PLOD2 and LOXL2 expression, in subsequent experiments we investigated the effects of activating these pathways individually or in combination using lung fibroblasts from patients with IPF. We selected the 48hr time point on the basis of our timecourse experiment (Figure 2), with all experiments performed using a minimum of 3 IPF cell lines. The effect of DMOG in the absence or presence of TGFβ1 upon PLOD2 and LOXL2 induction (Figure 4a-c) was comparable to that identified using normal control lung fibroblasts. When HIF stabilisation (DMOG or IOX2) and TGFβ1 were combined, a synergistic effect upon the induction of PLOD2 gene and protein expression was identified which was greater than either pathway alone (new Figure 4a and c). Whilst the gene expression of LOXL2 was also increased with the combination of HIF stabilisation and TGFβ1 a corresponding increase in protein expression within cell lysates was not apparent. As LOXL2 is processed intracellularly before being extracellularly secreted we therefore investigated whether increased secretion of LOXL2 was occurring, identifying that under conditions with HIF stabilisation in the absence or presence of TGFβ1, LOXL2 secretion was increased (new Figure 4d; new Figure 4—figure supplement 1).

Figure 5. To show a direct effect of PLOD2 and LOXL2 on pyridoline cross-linking and ECM stiffness, authors can do the same experiment as Figure 5 A and B with siRNA for PLOD2 and LOXL2 after IOX2 treatment and see whether this can revert the phenotype.

Please see the response to Essential Revision 3.

Figure 7. The claim of HIF/LOXL2/PLOD 2 colocalization needs to be supported by more images (quality and quantity). PLOD/LOXL2 constating is the same of Figure 2. Authors should provide co-staining of PLOD2/LOXL2 and HIF/CAIX in the same slide. If hard to obtain, PLOD2/LOXL2 with HIF and PLOD2/LOXL2 with CAIX costaining and show lower and higher magnification images and no few single events as in the actual figure.

Our findings were consistent across diagnostic tissue blocks and included independent histopathology review. We have included additional images (new Figure 7—figure supplement 1) demonstrating co-localisation of LOXL2/PLOD2 together with serial sections demonstrating the expression of CAIX and HIF1A.

Reviewer #3 (Recommendations for the authors):Overall this manuscript will be an important contribution to the field of lung fibrosis. The experiments are well executed and include adequate number of biological replicates for statistical analysis. One aspect I would consider modifying is the interpretation that the identified effect of HIF is acting in non-hypoxic conditions. At least from the data presented it doesn't look obvious that this is the case. If there are no compelling evidence for this claim I would modify the narrative and include this as a possibility in the discussion.

Thank you for this important comment. As described in our comments to the Editor, we now provide additional data demonstrating that selective FIH inhibition is sufficient to promote tissue stiffness.

Below are some points for improvement of the manuscript:Figure 1: The quantification of the correlation of expression of PLOD2 and LOXL2 is properly executed and include comparison with other cross-linking enzyme to show specificity of the result. Figure 1h corroborate this result by showing co-localization by ISH, and it would be helpful to add an high magnification of a negative region to show specificity of the staining.

As requested we have provided additional images including a high magnification of a negative region to show specificity of the staining (new Figure 1—figure supplement 1f).

Figure 7: Try to show co-localization from serial sections. This does not allows to determine if the same or adjacent cells were probed. A more definitive way to achieve this would be to perform ISH with multiple fluorescent probes, using the same technique (RNAscope) that here was used with a chromogenic approach that limit the co-localization to two transcripts. In alternative RNAscope + immunofluorescence could also be combined for the same purpose.

Thank you for this suggestion. We agree that fluorescent probes could be a helpful alternative, however here we have used IHC as our analysis pathway; this included formal histopathologist review for which the histopathologist requested this approach. We have included additional images (new Figure 7—figure supplement 1) demonstrating co-localisation of LOXL2/PLOD2 together with serial sections of CAIX and HIF1A.

The single cell data displayed in Fig8 could also be used to corroborate cellular co-expression of PLOD2/LOXL2 and HIF1a in addition to the signature score that are provided.

Thank you for this suggestion. As the HIF1a gene is constitutively expressed we felt that the HIF signature score would be an appropriate methodology to investigate this.

Figure 8: From the code description it seems that a dimensional reduction was created first, followed by dataset subsetting to include only 'Control' and 'IPF' in the final figure. Did the authors tried to create a dimensional reduction after subsetting the 'Kropski' dataset and see if this can improve the segregation of the clusters of interest?

Thank you for your suggestion. As suggested dimensional reduction was performed after subsetting the ‘Kropski’ dataset and we have now included this within the final figure and updated the uploaded code (new Figure 8; new Figure 8—figure supplement 1). Visually this did not improve segregation of the clusters of interest.

The color used in the tSNE plots in Figure 8a, 8c, S6a may be difficult to distinguish for color blind people. I suggest the use of alternative color palettes. For R, several packages are available for this purpose. The 'viridis' package for example use color blind accessible color scales.

Thank you for your suggestion. We have changed the plots to use an alternative color palette using the ‘viridis’ package as suggested.

The 'Cell Type' naming on top of the tSNE map is difficult to read, I suggest to adjust the position or to have it as a side legend.

As requested we have modified this to aid readability.